# AlphaBeta is not as good as you think: a simple class of synthetic games for a better analysis of deterministic game-solving algorithms

**Raphaël Boige**[*]     **Amine Boumaza**     **Bruno Scherrer**
Université de Lorraine, CNRS, Inria, LORIA, F-54000 Nancy, France

## Abstract

Deterministic game-solving algorithms are conventionally analyzed in the light of their average-case complexity against a distribution of random game-trees, where leaf values are independently sampled from a fixed distribution. This simplified model enables uncluttered mathematical analysis, revealing two key properties: root value distributions asymptotically collapse to a single fixed value for finite-valued trees, and all reasonable algorithms achieve global optimality. However, these findings are artifacts of the model's design: its long criticized independence assumption strips games of structural complexity, producing trivial instances where no algorithm faces meaningful challenges. To address this limitation, we introduce a class of synthetic games generated by a probabilistic model that incrementally constructs game-trees using a fixed level-wise conditional distribution. By enforcing ancestor dependencies, a critical structural feature of real-world games, our framework generates problems with adjustable difficulty while retaining some form of analytical tractability. For several algorithms, including AlphaBeta and Scout, we derive recursive formulas characterizing their average-case complexities under this model. These allow us to rigorously compare algorithms on deep game-trees, where Monte-Carlo simulations are no longer feasible. While asymptotically, all algorithms seem to converge to identical branching factor (a result analogous to that of independence-based models), deep finite trees reveal stark differences: AlphaBeta incurs a significantly larger constant multiplicative factor compared to algorithms like Scout, leading to a substantial practical slowdown. Our framework sheds new light on classical game-solving algorithms, offering rigorous evidence and analytical tools to advance the understanding of these methods under a richer, more challenging, and yet tractable model.

## 1 Introduction

In this work, we consider a class of deterministic two-player zero-sum games represented by trees of *height* $h$, where each node has a uniform *branching degree* $b$, as illustrated in Figure 1. Each level alternates between decision points for the maximizing and minimizing players (with the root always being a max node). Internal nodes propagate values from their children via alternating $\min/\max$ operators, reflecting optimal play. Clearly, the entire tree is determined by its leaf values, and solving it involves recursively applying $\min$ and $\max$ operations until the root value is resolved.

Game-solving algorithms are conventionally evaluated [1, 4, 8] by the number of leaf evaluations required. For example, brute-force search evaluates all $b^h$ leaves, corresponding to a *branching factor* of $b$—the average nodes evaluated per level, see Section 2 for formal definitions. Notably, even with prior knowledge of the root value, verifying it requires evaluating at least one node per

---

[*]Corresponding author: {name}.{surname}@inria.fr

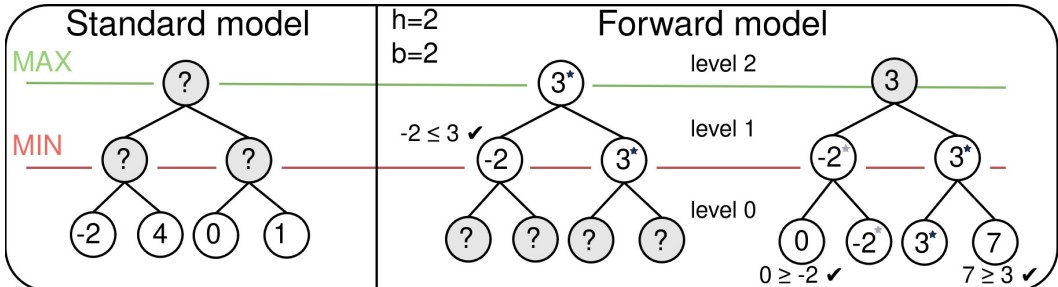

Figure 1: Illustration of a game-tree of height $h = 2$ and branching degree $b = 2$. (Left) In the *standard model*, leaf values are independently sampled from a distribution. (Right) In the *forward model*, intermediate values are sampled progressively, level by level, until leaf nodes are reached. One random child is chosen to inherit the root value, illustrated by a star symbol, and the remaining ones are sampled according to a fixed distribution, truncated to respect the minimax constraints.

max level and all $b$ nodes per min level. This results in a complexity of $b^{\frac{h}{2}}$, or a branching factor of $\sqrt{b}$, establishing global upper and lower bounds for all algorithms. ALPHA-BETA, as shown in [2], achieves these bounds under optimally ordered worst-case and best-case trees. This has motivated the emergence of average-case analyses, which aim to understand how game-solving algorithms perform on a diverse collection of random trees. The classical approach, hereafter called the *standard model*[1], samples leaf values independently from a fixed distribution. While mathematically tractable, this model exhibits critical flaws: when leaf values are restricted to a finite set, Pearl [5] proved that the root value asymptotically collapses to a single fixed value for all distributions. This collapse renders algorithm comparisons questionable: instances become trivial, as methods merely confirm a predetermined value shared among trees.

A practical consequence is the *standard model*'s assignment of global optimality to multiple algorithms. ALPHA-BETA, for instance, provably achieves the $\sqrt{b}$ branching factor asymptotically when $h$ and $b$ tend to infinity, except for one rare case discussed later. This is remarkable, since no alternative algorithm, even in principle, can outperform them asymptotically. While this has been interpreted as a sign of algorithmic maturity, we believe instead that it reflects deficiencies in the evaluation framework. When trees homogenize to trivial instances with a single root value, comparisons lose meaning as algorithms face no substantive challenges to distinguish their performance. Previous works have pinpointed that these limitations stem from the model's independence assumption [6, 7, 2], which can't model dependencies between sibling nodes. Real-world games like chess possess an important type of ancestor dependencies: correlation among sibling nodes. In average a winning position for the white player should have many wins (and few losses) in the subtree emanating from it. Crucially, the independence assumption eliminates such complexity, rendering the *standard model* a poor proxy for practical scenarios, and a poor benchmark for comparing algorithms.

In this work, we specify and analyze a synthetic game-tree model: the *forward model*[2], that addresses these limitations. By constructing trees level-by-level with a conditional distribution that enforces ancestor dependencies, our approach captures two critical properties: (1) sibling values correlate conditioned on their parent value, coarsely mimicking the strategic continuity of real games, and (2) game difficulty can be better modulated, enabling more challenging benchmarks for algorithm analysis. In Section 3 we further detail our *forward model*. Within this framework, we characterize the behavior of classic algorithms with recursive formulas for their average-case complexity, including ALPHA-BETA, which we develop in Section 5. These formulas are convenient for asymptotic analysis as well as limit-depth evaluations, as they allow simulating the behavior of algorithms on deep trees much more efficiently than Monte-Carlo simulations. Our findings reveal that in the asymptotic regime, all algorithms share the same branching factor, as a function of the distribution chosen. Theoretically, this suggests that there is little reason to prefer one algorithm over another. However, finite-depth analysis, in Section 6, reveals critical practical differences masked by asymptotics. Specifically, ALPHA-BETA incurs a larger multiplicative constant, and needs in average to evaluated more leaf nodes compared to other algorithms like SCOUT. Finally, to allow reproduction of the numerical results presented

---

[1]The *standard model* is often referred to as the Pearl-game or P-game model in the literature.

[2]In the rest of this work, we freely use the terms *Forward-game* or *F-game* to designate a game generated by the *forward model*.

in this paper, we open-source our codebase which permits the precise computation of average-case complexities of several algorithms and for game-trees up to height $h \approx 5000$[3].

## 2 Notations and background

In this section, we formalize the framework for analyzing deterministic game-solving algorithms, focusing on average-case complexity under a probabilistic tree generation model.

**Game formalism** We model games as complete $b$-ary trees of *height* $h$, where leaf nodes hold values from a space $\mathcal{V}$. Formally, a minimax node value $V_h$ with children values $(V_{h-1}^i)_{i \leq b}$ can be written $V_h = \max_{i \leq b}(V_{h-1}^i)$ (*resp.* $\min_{i \leq b}(V_{h-1}^i)$) if level $h$ is at an even (*resp.* odd) distance from the root. However, in two-player zero-sum games, by using the identity $\min(a, b) = -\max(-a, -b)$, we can formulate an equivalent negamax view where the alternate min/max are replaced by a single operator. This defines a negamax node value $W_h$ with children values $(W_{h-1}^i)_{i \leq b}$ as:
$$W_h = \max_{i \leq b}(-W_{h-1}^i) \tag{1}$$
We can see the two views are equivalent, as $W_h = V_h$ (*resp.* $-V_h$) if level $h$ is at an even (*resp.* odd) distance from the root. In the rest of this work, we use the negamax view, as it often simplifies algorithmic descriptions and formal analysis, conveniently halving the number of cases to study.

**Algorithmic complexity and branching factor** To measure the efficiency of a deterministic algorithm $A$, the main value of interest is its *average-case complexity $I_A(h)$*, *i.e.* the expected number—over a distribution of trees generated randomly—of *leaf node* inspections required by the algorithm $A$ to terminate.

The *asymptotic branching factor $r_A$* quantifies the complexity growth of the algorithm with the height of the tree:
$$r_A = \lim_{h \to \infty} \sqrt[h]{I_A(h)}. \tag{2}$$
This represents the effective branching rate per level. As stated before, global lower and upper bounds are known for these quantities, for all algorithms $A$: $b^{\frac{h}{2}} \leq I_A(h) \leq b^h$ and $\sqrt{b} \leq r_A \leq b$.

## 3 The forward model

This section formally introduces the *forward model*: a simple synthetic game model that recursively generates game-trees, down from the root node, through a level-wise sampling process that enforces the negamax constraint from Equation 1 at each step. This model resembles previous synthetic game models that generate trees in a top-down fashion [17, 20]. Still, to the best of our knowledge, this work provides the first analysis of this conceptually simple yet rich game model. The sampling process can be described as follows: starting from the root node, assuming it holds value $x$, one of its $b$ children is uniformly selected to inherit the parent's negated value $-x$, ensuring compliance with the negamax constraint. The remaining $(b-1)$ children are then sampled from the level-wise distribution $\mu$, with support dynamically truncated conditionally on $x$. For instance, if $\mu$'s initial support is $\{-n, ..., n\}$ for an integer $n$, it gets truncated (and re-normalized) to $\{-x, ..., n\}$. This procedure, dubbed FORWARD-SAMPLE and formally described in Algorithm 1, is first called on the root node, and recursively applied to each node until all leaves have been generated. For simplicity, we choose the same distribution $\mu$ for each node, and we draw the root value of the tree according to $\mu$ as well.

---

**Algorithm 1:** FORWARD-SAMPLE($x,h,b$)

---

**Input:** Node value $x$, tree height $h$,
      branching degree $b$.
**Output:** Children values list.
**if** $h = 0$ **then return** $[\,]$   // Leaf nodes
$x'_{list} \leftarrow [\,]$
$k_{sample} \sim \mathcal{U}\{1, b\}$
**for** *k=1...b* **do**
    **if** $k = k_{sample}$ **then** $x' = -x$
    **else** $x' \sim \mu(\cdot \mid \cdot \geq -x)$
    $x'_{list}.append(x')$
**end**
**return** $x'_{list}$

---

Unlike traditional models that first assign leaf values and propagate them backward through min/max rules, our approach builds values progressively from the root downward, hence the "forward" designation. As demonstrated in the next section, this property simplifies the complexity analysis for game-solving algorithms, as intermediate outcomes are conditionally known during tree construction.

---

[3]Link to the repository: https://github.com/Egiob/alphabeta

# 4 A binary-valued example: the analysis of the SOLVE algorithm

In this section, we consider binary-valued trees that represent two-outcome games, loss or win (i.e., $\mathcal{V} = \{0, 1\}$). We focus on SOLVE, a canonical algorithm for two-outcome games, and contrast its behavior under the *standard model* versus our *forward model*. Key derivations appear in Appendix C.

**Algorithm description and standard model limitations**  The SOLVE algorithm (pseudo-code in Appendix C) determines win/loss (1/0) outcomes by iteratively scanning through children and evaluating them until an opponent loss (0) is found, in this case it early stops and returns a win (1), if none is found, it returns a loss (0). Under the *standard model*, with probability $q_0$ of drawing a 0 for a leaf node, SOLVE almost always achieves a globally optimal branching factor:

$$r_{\text{SOLVE}}^{\text{STANDARD}} = \begin{cases} \sqrt{b} & \text{if } q_0 \neq 1 - \xi_b \text{ (very easy instances)} \\ \frac{\xi_b}{(1 - \xi_b)} = \mathcal{O}(\frac{b}{\log b}) & \text{if } q_0 = 1 - \xi_b \text{ (special case)} \end{cases}, \tag{3}$$

where $\xi_b$ is the positive root of $x^b + x - 1 = 0$ [5]. Since only this exceptional regime generates non-trivial instances, it has been thoroughly analyzed in the literature [4, 5]. However, even this hardest regime generates much easier instances than those of an ordering-invariant worst-case model [13]:

$$r_{\text{SOLVE}}^{\text{RAND-WC}} = \frac{b - 1 + \sqrt{b^2 + 14b + 1}}{4} = \mathcal{O}(b). \tag{4}$$

The *standard model*'s abrupt transition between $\sqrt{b}$ and $\log b/b$ regimes, described in Equation 3, reveals its inability to generate smoothly tunable or maximally hard instances.

**Complexity analysis**  Under the *forward model*, where $\mu = \mathcal{B}(q)$ (Bernoulli distribution with $q$ the probability of drawing a 0), we derive (Appendix C) a closed-form expression for SOLVE's branching factor:

$$r_{\text{SOLVE}} = \frac{t(q, b) + \sqrt{t(q, b)^2 + 4b}}{2}, \quad t(q, b) = \sum_{k=1}^{b-1} \frac{1 + (b - k - 1)q}{b} k(1 - q)^k. \tag{5}$$

If $q = 1$, the model collapses to a trivial tree (alternating levels full of zeros and full of ones) leading to a $\sqrt{b}$ branching factor. However, if $q = 0$, it matches exactly the worst-case complexity of Equation 4. For values of $q$ in-between, the monotonicity of $r_{\text{SOLVE}}$ with respect to $q$ guarantees that any branching factor from easiest to hardest case can be reached, allowing adjustable difficulty calibration (unattainable under the *standard model*). In particular, a continuous range of $q$ values leads to asymptotically hardest instances as stated in the following theorem, the proof of which is in Appendix A.1.

**Theorem 4.1.**  *For $b \in \mathbb{N}$ and for all $q \in [0, \frac{1}{b}]$, the branching factor of SOLVE satisfies $r_{\text{SOLVE}} = \mathcal{O}(b)$.*

In the next section, we analyze classical algorithms on a more general type of trees under our original *forward* tree model.

# 5 Average-case analysis of classic algorithms

Even though binary-valued games offer a simplified analysis, they cannot reflect the diversity of real games, which are best modeled using a broader value range. In this section, we analyze the algorithms TEST, ALPHA-BETA and SCOUT, on trees with values in $\{-n, \ldots, n\}$, *i.e.* $\mu$ is a categorical distribution $Cat(p_{-n}, \ldots, p_n)$.

## 5.1 Analysis of TEST

**Algorithm description**  Described in Algorithm 2, TEST answers, given a threshold $s$, whether the root value $x$ satisfies $x \geq s$. Like SOLVE, TEST iteratively evaluates every child node by calling a negated version of itself and terminates early whenever it finds a value validating the condition $x \geq s$. It is almost identical to applying SOLVE to a binarized version of the same tree where leaves $l_i$ are converted to 1 if $l_i \geq s$ and 0 otherwise—with the main difference being TEST not only returns the

---

**Algorithm 2:** TEST($N, h, s$), *negamax* form

**Input:** Current node $N$, search depth $h$, threshold $s$

**Output:** Certificate value determining if $N$'s value is greater or equal than $s$

**if** $h = 0$ **then return** $N.value$
$best \leftarrow -\infty$
**foreach** $N'$ *in N.children* **do**
  $value \leftarrow -\text{TEST}(N', h - 1, -s + 1)$
  $best \leftarrow \max(best, value)$
  **if** $best \geq s$ **then break**
**end**
**return** $best$

---

binary result of the assertion but also returns a certificate value determining whether $x \geq s$ or not. This makes it a useful building block for other game-solving algorithms. For instance, a simple algorithm could brute-force over all possible thresholds $s$ to identify $x$, note that this approach can be optimized via bisection. In Section 6, we compare these TEST-BRUTEFORCE and TEST-BISECTION approaches to the ALPHA-BETA and SCOUT algorithms. In the *standard model*, if leaf values are drawn according to a distribution $\nu_0$ with cumulative distribution $F_{\nu_0}$, then:

$$r_{\text{TEST}}^{\text{STANDARD}}(\nu_0, s) = r_{\text{SOLVE}}^{\text{STANDARD}}(q_0 = F_{\nu_0}(s)). \tag{6}$$

This equivalence suggests that games with discrete values are not fundamentally asymptotically harder than games with binary values. In the following we characterize TEST under the *forward model* and show that its branching factor coincides with that of SOLVE on a worst-case distribution.

**Complexity analysis** For a given threshold value $s \in \{-n+1, \dots, n\}$ we are interested in the average-complexity of TEST$(s)$ defined as $I_{\text{TEST}}^s(h) = \mathbb{E}_{X \sim \mu}[I_{\text{TEST}}^{X,s}(h)]$, where $I_{\text{TEST}}^{x,s}(h)$ denotes the expected complexity of TEST when the root value is $x \in \{-n, \dots, n\}$. To model intermediate evaluation states, we extend this definition to $I_{\text{TEST}}^{x,s}(h, c)$, representing the complexity when the current node (with value $x$) has $c \leq b$ remaining children to evaluate (all other nodes still have $b$ children). Thus, the base case satisfies: $I_{\text{TEST}}^{x,s}(h) = I_{\text{TEST}}^{x,s}(h, c = b)$. A necessary tool for expressing $I_{\text{TEST}}^{x,s}(h)$ is the auxiliary function $J_{\text{TEST}}^{x,s}(h, c)$, which represents the same complexity value, but conditioned on the probabilistic event that the "special child" (inheriting the root's value, enforced by the negamax constraint) has already been identified. A recursive system characterizing the average-case dynamics of TEST can be derived by analyzing its execution flow under the *forward model*. Upon evaluating the first child ($c = b$): (1) with probability $1/c$, TEST encounters the "special child", inheriting the root value $-x$. The algorithm must then fully evaluate this child node by a recursive call to a negated version of TEST at height $h-1$, if no cutoff occurs (*i.e.* $x < s$) it continues evaluating the root node (height $h$), but with only $c-1$ remaining children, and knowing the "special child" has been found (cost of $J_{\text{TEST}}^{x,s}(h, c-1)$ instead of $I_{\text{TEST}}^{x,s}(h, c-1)$); (2) with probability $(c-1)/c$, TEST encounters a "normal" child whose value $X'$ is sampled from $\mu$ truncated to $\{-x, \dots, n\}$ and in the absence of cutoff (*i.e.* $-X' < s$), the algorithm proceeds to evaluate the remaining children of the root node (cost of $I_{\text{TEST}}^{x,s}(h, c-1)$). This gives the following equation for $I_{\text{TEST}}^{x,s}$:

$$
\begin{aligned}
I_{\text{TEST}}^{x,s}(h, c) = \frac{1}{c} & \overbrace{\left[ I_{\text{TEST}}^{-x, -s+1}(h-1, b) + \mathbb{1}_{\{x < s\}} J_{\text{TEST}}^{x,s}(h, c-1) \right]}^{\text{Special child } X' = -x} \\
+ \frac{c-1}{c} & \underset{\substack{X' \sim \mu \\ X' \geq -x}}{\mathbb{E}} \underbrace{\left[ I_{\text{TEST}}^{X', -s+1}(h-1, b) + \mathbb{1}_{\{-X' < s\}} I_{\text{TEST}}^{x,s}(h, c-1) \right]}_{\text{Normal child } X' \sim \mu}.
\end{aligned}
\tag{7}
$$

The auxiliary function $J_{\text{TEST}}^{x,s}(h, c)$ follows the same logic, but being conditioned on the event "the special child already has been found", it only allows one of the branches, so the equation simplifies to:

$$J_{\text{TEST}}^{x,s}(h, c) = \underset{\substack{X' \sim \mu \\ X' \geq -x}}{\mathbb{E}} \left[ I_{\text{TEST}}^{X', -s+1}(h-1, b) + \mathbb{1}_{\{-X' < s\}} J_{\text{TEST}}^{x,s}(h, c-1) \right]. \tag{8}$$

Note that for the end case $h = 0$, both $I$ and $J$ equal 1 (a tree with only one node incurs a cost of 1), and for $c = 0$, both $I$ and $J$ equal 0 (because no remaining child incurs no additional costs). Equations 7 and 8, characterizing the complexity of TEST—as well as Equations 9 and 10 (resp. Equations 11 and 12) characterizing the complexity of ALPHA-BETA (resp. SCOUT)—were numerically validated through an extensive comparison with Monte-Carlo simulations, see Appendix B.

Conveniently, the intrinsic linear nature of Equations 7 and 8 makes it possible to write the system in matrix form. This facilitates the efficient numerical computation of the complexity of TEST$(s)$ by matrix iteration and that of its branching factor as the spectral radius (eigenvalue with highest magnitude) of this matrix. Additionally, we define a global branching factor for the TEST algorithm, corresponding to the complexity of the average TEST, or equivalently, the complexity of the hardest TEST (over all threshold values $s$): $r_{\text{TEST}} = \max_s r_{\text{TEST}}(s)$. Interestingly, against a worst-case distribution $\mu = \delta_n$ (all probability mass concentrated on $n$) $r_{\text{TEST}}$ exactly attains the bound of Equation 4. As a consequence, following the conclusions from the analysis of SOLVE in Section 4, the *F-games* are typically harder problems than the games generated under *standard model*, even for discrete-valued trees. That makes $r_{\text{TEST}}$ an interesting and easy-to-compute quantity to gauge the difficulty induced by the choice of a distribution $\mu$. We use this property in Section 6 as a measure of difficulty: if $r_{\text{TEST}}$

is large (resp. small) the game-trees are considered hard (resp. easy). In the following section, we characterize the average-case complexity of ALPHA-BETA and compare its branching factor to that of TEST.

## 5.2 Analysis of ALPHA-BETA

**Algorithm description** The ALPHA-BETA algorithm improves upon classical full negamax search by pruning branches that cannot influence the root value. It tracks two evolving bounds: $\alpha$ (the worst-case guarantee for the maximizing player) and $\beta$ (the best-case allowance for the minimizing player). As described in Algorithm 3, upon a child evaluation, achieved through a recursive call with negated parameters $\alpha' = -\beta$ and $\beta' = -\alpha$, ALPHA-BETA updates the current best value and early terminates whenever it exceeds $\beta$,

---

**Algorithm 3:** ALPHABETA$(N, h, \alpha, \beta)$, *negamax* form

---

**Input:** Current node $N$, search depth $h$, lower-bound $\alpha$, upper-bound $\beta$
**Output:** Value of node $N$.
**if** $h = 0$ **then return** $N.value$
$best \leftarrow -\infty$
**foreach** $N'$ in $N.children$ **do**
    $value \leftarrow -\text{ALPHABETA}(N', h - 1, -\beta, -\alpha)$
    $best \leftarrow \max(best, value)$
    **if** $best \geq \beta$ **then break**
    $\alpha \leftarrow \max(\alpha, best)$
**end**
**return** $best$

---

and it potentially updates $\alpha$ for next sibling evaluation. While typically invoked with a full-window ($\alpha = -\infty$, $\beta = +\infty$) to compute the root value $x$ exactly, ALPHA-BETA can also operate with bounded intervals, in this case if $x$ is not comprised in $[\alpha, \beta]$, it will return a certificate value (like TEST) asserting whether $x \geq \beta$ or $x \leq \alpha$. A well-known [14] connection to TEST appears with a so-called *null-window* ($\alpha = s - 1$, $\beta = s$), in this case, ALPHA-BETA becomes functionally equivalent to TEST($s$), incurring the same complexity and producing identical certificates.

**Complexity analysis** We define $I_{\text{AB}}^{x,\alpha,\beta}(h, c)$ the average-case complexity of ALPHA-BETA called with parameters $\alpha < \beta$ and using same notations as before. We are interested in $I_{\text{AB}}(h) = \mathbb{E}_{X \sim \mu}[I_{\text{AB}}^{X,-n,n}(h)]$, the average complexity of ALPHA-BETA called with a full-window, and in $r_{\text{AB}}$, its branching factor. The derivation of this complexity follows closely the one for TEST and yields a similar system of recursive equations. For a node with value $X'$, where ALPHA-BETA is called with parameters $\alpha$ and $\beta$, the main differences with TEST are: (1) the cutoff condition now becomes $-X' < \beta$, (2) the recursive calls to ALPHA-BETA are made with parameters $-\beta$ and $-\alpha$ and (3) unlike in TEST, further calls at the same level may use an updated value of $\alpha$ if the current child has the best value encountered so far. Assuming $J_{\text{AB}}$ follows a similar definition to that of $J_{\text{TEST}}$:

$$I_{\text{AB}}^{x,\alpha,\beta}(h, c) = \frac{1}{c}\left[I_{\text{AB}}^{-x,-\beta,-\alpha}(h - 1, b) + \mathbb{1}_{\{x < \beta\}} J_{\text{AB}}^{x,\max(\alpha,x),\beta}(h, c - 1)\right]$$
$$+ \frac{c - 1}{c} \mathop{\mathbb{E}}_{\substack{X' \sim \mu \\ X' \geq -x}}\left[I_{\text{AB}}^{X',-\beta,-\alpha}(h - 1, b) + \mathbb{1}_{\{-X' < \beta\}} I_{\text{AB}}^{x,\max(\alpha,-X'),\beta}(h, c - 1)\right], \quad (9)$$

and $J_{\text{AB}}^{x,\alpha,\beta}(h, c) = \mathop{\mathbb{E}}_{\substack{X' \sim \mu \\ X' \geq -x}}\left[I_{\text{AB}}^{X',-\beta,-\alpha}(h - 1, b) + \mathbb{1}_{\{-X' < \beta\}} J_{\text{AB}}^{x,\max(\alpha,-X'),\beta}(h, c - 1)\right]. \quad (10)$

This system of equations resembles Equations 7 and 8 and allows us to run comprehensive numerical simulations, which lead to the remarkable observation that TEST and ALPHA-BETA share the same branching factor. We find that the equality of the branching factor holds theoretically, as indicated in the following theorem (proof in Appendix A.2):

**Theorem 5.1.** ALPHA-BETA *called with a full window* $\{-n, ..., n\}$ *is more efficient than the approach consisting in testing every possible value with the* TEST *procedure (i.e.,* TEST-BRUTEFORCE*) and* ALPHA-BETA *and* TEST *share the same asymptotic branching factor, in the precise sense that for* $h \geq 0$ *and* $x \in \{-n, \ldots, n\}$:

$$I_{\text{AB}}^{x,-n,n}(h) \leq \sum_{s=-n+1}^{n} I_{\text{TEST}}^{x,s}(h) \quad and \quad r_{\text{AB}} = r_{\text{TEST}}.$$

This result shows that there is no asymptotic gain of using ALPHA-BETA over a simple TEST-BRUTEFORCE approach, that calls TEST $2n$ times. Moreover, numerical results, presented in Section 6 suggest that these two algorithms present a deeper identical behavior: both for the asymptotic limit and the

multiplicative constant characterizing the convergence rate. This result is quite remarkable, since the question answered by ALPHA-BETA—determining the precise value of the game—intuitively seems to be much harder than the question answered by TEST, that only solves a binary problem. In the next section, we conduct a similar average-case analysis of the SCOUT algorithm.

## 5.3 Analysis of SCOUT

**Algorithm description** SCOUT incorporates the TEST algorithm into a procedure similar to ALPHA-BETA, described in Algorithm 4. Before evaluating any node, SCOUT first performs a call to TEST to check whether the child's value strictly exceeds $\alpha$. Only if this test returns true, indicating potential for improvement, does SCOUT proceed to evaluate the node in full, by a negated recursive call to itself, and updates $\alpha$ to the current (higher) value. Like ALPHA-BETA, it is most often called with $\alpha = -\infty$, but can also be called with any value of $\alpha$ and potentially a parameter $\beta$ too (triggering a cutoff whenever $\alpha \geq \beta$). At first glance, this approach may appear inefficient:

---

**Algorithm 4:** SCOUT($N, h, \alpha, \beta$)

**Input:** Current node $N$, search depth $h$,
  lower-bound $\alpha$, upper-bound $\beta$
**Output:** Value of node $N$
**if** $h = 0$ **then return** $N.value$
**if** $\alpha \geq \beta$ **then return** $\alpha$
**foreach** $N'$ *in N.children* **do**
  $test \leftarrow -\text{TEST}(N', h-1, -\alpha)$
  **if** $test > \alpha$ **then**
    $\alpha \leftarrow -\text{SCOUT}(N', h-1, -\beta, -\alpha-1)^4$
  **end**
  **if** $\alpha \geq \beta$ **then break**
**end**
**return** $\alpha$

---

when a TEST returns true, subsequent evaluations revisit some leaf nodes already examined during the threshold check. However, previous experimental results [9, 12] suggest that the waste incurred by SCOUT's reevaluation of some nodes is not substantial. Furthermore, improved variants of SCOUT—Principal Variation Search (PVS) and NegaScout [10, 9]—are still used in modern game engines [24]. In this work we focus on the original SCOUT algorithm, which has a simpler formal analysis: unlike its improved variants, it does not use the certificate value $v$ but only the boolean outcome $\mathbb{1}_{v>\alpha}$ of the TEST procedure. We believe that PVS and NegaScout improvements should not change the asymptotic behavior of SCOUT, and we leave their formal analysis for future work.

**Complexity analysis** Mirroring the ALPHA-BETA analysis, we define $I^{x,\alpha,\beta}_{\text{SCOUT}(h,c)}$ as the complexity of SCOUT conditioned on $x$ and $\alpha < \beta$ and we are interested in the complexity and branching factor for a full-window $I_{\text{SCOUT}(h)} = \mathbb{E}_{X' \sim \mu}[I^{X',-n,n}_{\text{SCOUT}(h)}]$ and $r_{\text{SCOUT}}$. We can write the following recursive equations:

$$I^{x,\alpha,\beta}_{\text{SCOUT}(h,c)} = \frac{1}{c}\left[ I^{-x,-\alpha}_{\text{TEST}(h-1,b)} + \mathbb{1}_{\{\alpha<x\}} I^{-x,-\beta,-\alpha-1}_{\text{SCOUT}(h-1,b)} + \mathbb{1}_{\{x<\beta\}} J^{x,\max(\alpha,x),\beta}_{\text{SCOUT}(h,c-1)} \right] +$$

$$\frac{c-1}{c}\mathbb{E}_{X'\geq-x}\left[ I^{X',-\alpha}_{\text{TEST}(h-1,b)} + \mathbb{1}_{\{\alpha<-X'\}} I^{X',-\beta,-\alpha-1}_{\text{SCOUT}(h-1,b)} + \mathbb{1}_{\{-X'<\beta\}} I^{x,\max(\alpha,-X'),\beta}_{\text{SCOUT}(h,c-1)} \right], \tag{11}$$

$$J^{x,\alpha,\beta}_{\text{SCOUT}(h,c)} = \mathbb{E}_{X'\geq-x}\left[ I^{X',-\alpha}_{\text{TEST}(h-1,b)} + \mathbb{1}_{\{\alpha<-X'\}} I^{X',-\beta,-\alpha-1}_{\text{SCOUT}(h-1,b)} + \mathbb{1}_{\{-X'<\beta\}} J^{x,\max(\alpha,-X'),\beta}_{\text{SCOUT}(h,c-1)} \right] \tag{12}$$

This defines a system very similar to that of ALPHA-BETA, and extensive numerical studies that we conducted suggest that the branching factor of SCOUT coincides with that of TEST and ALPHA-BETA. We state here an analogous result to the one we obtained for ALPHA-BETA (proof in Appendix A.3):

**Theorem 5.2.** *SCOUT called with a full window $\{-n, ..., n\}$ is more efficient than the approach consisting in testing every possible value with the TEST procedure (i.e., TEST-BRUTEFORCE). For $h \geq 0$ and $x \in \{-n, ..., n\}$:*

$$I^{x,-n,n}_{SCOUT}(h) \leq \sum_{s=-n+1}^{n} I^{x,s}_{\text{TEST}}(h) \quad and \quad r_{SCOUT} \leq r_{\text{TEST}}.$$

This result is weaker than Theorem 5.1, in the sense that we did not manage to prove that SCOUT is asymptotically equivalent to TEST. The missing part is to show that $r_{\text{SCOUT}} \geq r_{\text{TEST}}$, which is suggested by numerical simulations.

In the next section, we experimentally compare all presented algorithms for deep trees and different parametrizations of the *forward model*.

---

[4]Though this is not standard in the description of SCOUT, it seems that we can freely increment $\alpha$ by one each time a strict test is proven true, because if $x > \alpha$ then $x \in [\alpha + 1, \beta]$, facilitating our formal analysis of SCOUT.

# 6 Finite-depth numerical analysis

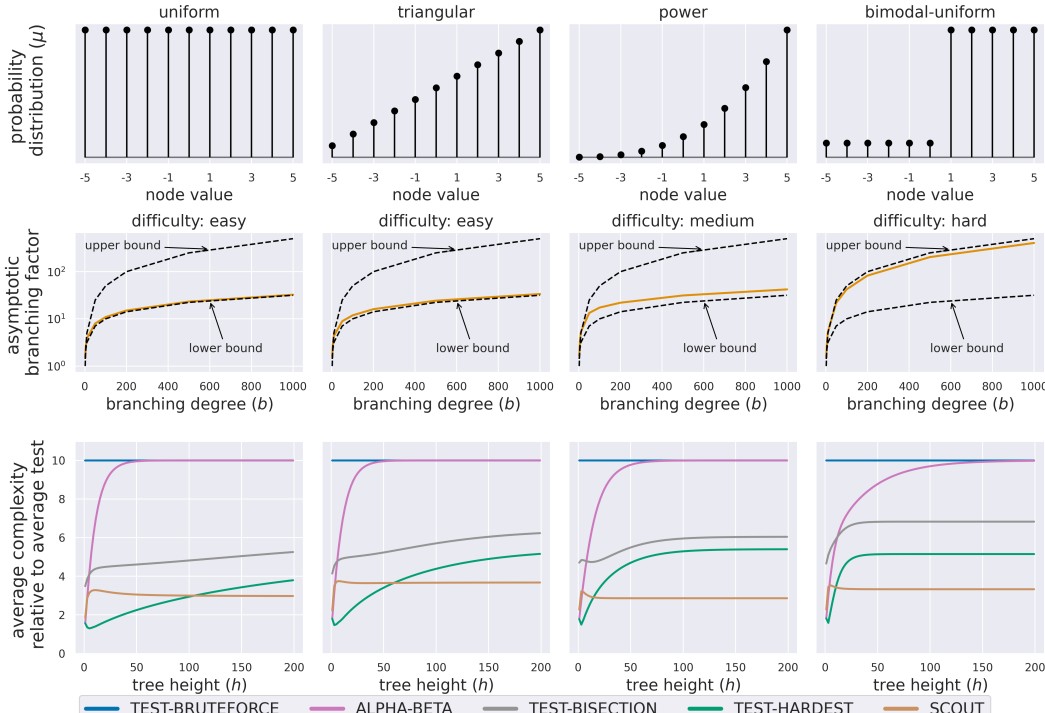

Figure 2: Finite-depth comparison of average-complexities. Each column corresponds to a parametrization of the *forward model*, consisting in the choice of the distribution $\mu$. (Top) Probability mass function of $\mu$ (Middle) Difficulty of the generated instances induced by the choice of $\mu$, measured as the dependency of the branching factor $r$ to the branching degree $b$. (Bottom) Relative average-complexity w.r.t. the average performance of the algorithm TEST, this allows us to visualize the sub-exponential multiplicative factor as a function of the height $h$ (with $b = 10$, $n = 5$). For each parametrization ALPHA-BETA demonstrates the same convergence rate as TEST-BRUTEFORCE. Conversely, SCOUT and TEST-BISECTION exhibit faster convergence, and remarkably SCOUT even seems to incur a smaller multiplicative constant than TEST-HARDEST.

In Section 5 we have established that TEST and ALPHA-BETA shared the same asymptotic branching factor, which also coincides with that of SCOUT numerically. To gain more insights into the behavior of these algorithms, and especially better understand their sub-exponential convergence rates, we conduct a finite-depth experimental analysis, using the recursive equations derived earlier in this work.

**Baselines** In addition to ALPHA-BETA and SCOUT, we choose to consider two TEST-based baselines: (1) TEST-BRUTEFORCE, which comprehensively applies TEST for every threshold $s \in \{-n + 1, \ldots, n\}$ and (2) TEST-BISECTION, which uses a binary search approach to reduce the number of TEST trials from $\mathcal{O}(n)$ to $\mathcal{O}(\log n)$. We also introduce the TEST-HARDEST baseline which corresponds to the most expensive call to TEST across all threshold $s$ values.

**Parametrization** To best compare the algorithms, we design diverse instantiations of the *forward model*, with different distributions $\mu$. Ideally, we want these parametrizations to cover a wide range of problem difficulty, from the simplest to the hardest case. We suggest that the asymptotic branching factor (common to all algorithms) $r$ can be interpreted as a measure of the intrinsic game's difficulty. We propose an arbitrary and informal way of classifying the difficulty: for a given distribution $\mu$, value range $n$ and branching degree $b$, if the branching factor $r$ is close to $\sqrt{b}$ (the lower bound) then we classify it as easy, similarly if it's close to $b$ (the upper bound) we classify it as hard, otherwise we classify it as medium. For the specific choice of these distributions we start with a uniform distribution—a natural choice that maximizes the diversity (entropy) of root values. Then, we choose distributions that assign increasing probability mass on positive atoms, because we have found empirically that this leads to produce problems of increased difficulty. The intuition is that a node with value $x$ will draw children in $\{-x, \ldots, n\}$, thus the larger $x$, the wider the interval, and the higher

the diversity of node values in the generated tree. As displayed in the top row of Figure 2, we choose a triangular distribution, a power-law (cubic) distribution and finally a bimodal-uniform distribution with more than $1/b$ mass concentrated on positive atoms (following a criterion similar to that of Theorem 4.1).

**Results**   We present in Figure 2, the results of the finite-depth average-case complexity comparison. Each column in the figure represents, top to bottom, the probability distribution $\mu$ used, the difficulty this choice induces in the generated game-trees and finally the actual performance of the compared algorithms. To best compare the relative performance of algorithms, we divide the average-complexity by that of TEST, computed as an average of all possible values of threshold $s$. This allows us to easily distinguish the algorithm transient regimes by focusing on the sub-exponential multiplicative factor and ignoring the mechanical effect of the complexity increasing with the height of the tree. A striking result appears: ALPHA-BETA consistently ends up as the worst algorithm across all evaluation setups as well as for every tree size. Remarkably, it seems to mirror the performance of the naive TEST-BRUTEFORCE baseline for very deep trees. It may suggest that through recursive calls, the $\alpha$ and $\beta$ parameters of ALPHA-BETA rapidly reduce to null-window situations where $\alpha = \beta - 1$, and end up comprehensively testing all possible threshold values, thus becoming equivalent to the TEST-BRUTEFORCE approach. This might be a symptom of a deeper asymptotic equivalence between ALPHA-BETA and TEST-BRUTEFORCE, which seem to behave identically asymptotically as well as in their sub-exponential constant factor.

Conversely, SCOUT seems to consistently outperform ALPHA-BETA and to achieve the best performance for every game difficulty. Unlike the conclusions drawn in a fixed-depth analysis under the *standard model* [5], it supports the practical superiority of SCOUT over ALPHA-BETA as hinted in multiple numerical studies [9, 12]. TEST-BISECTION also displays a strong performance, we believe this comes from its conceptual similarity to the MTD(f) algorithm, whose practical superiority over ALPHA-BETA have been suggested in the past [16]. Interestingly enough, SCOUT seems to even outperform TEST-HARDEST for deeper trees. This result is counter-intuitive since SCOUT's cost mostly comes from calls to TEST. This probably suggests that SCOUT behaves like an adaptive version of TEST, which updates the threshold value according to the values encountered, unlike TEST, which has a fixed threshold value, making it more sensitive to worst-case situations.

## 7   Related works

The average-case analysis of minimax algorithms originated from studying ALPHA-BETA under the *standard model* with independent leaf values [1, 4], where it achieves optimality [8, 11]. Subsequent algorithms, SCOUT [5], MTD(f) [14] and PVS (sometimes called NegaScout) [10] were proven asymptotically equivalent. Critiques of the *standard model* highlight its unrealistic independence assumption [2, 7]. Alternative models introduce ancestor dependencies, e.g., the *incremental model* maintains a heuristic value at each node, and defines leaf values as the sum of values along the path to the root node. Only limited settings of this formulation of the *incremental model* have been successfully analyzed [15, 3] and a more general analysis is yet to be proven feasible. Prior works also explore the idea of forward tree generation [17, 20], where values are generated top-down like in our *forward model*. The closest model to ours is the Prefix Value Game Tree Model [20], it is an instantiation of the *incremental model* where heuristic values of nodes are computed additively from their parents, and one child is always attributed a zero increment, this resembles our *forward model* where one child inherits the value of the parent. However, to our knowledge, this model has not yet been successfully theoretically analyzed before. Moreover, we are confident that the analysis techniques we developed in this work could be very simply adapted to it.

Another popular approach in modern game solving is Monte-Carlo Tree Search (MCTS) [18, 21], however, a common theoretical average-complexity analysis under a shared framework with minimax-based game-solving algorithms is currently missing. This is explained by technical reasons. MCTS algorithms only offer asymptotic convergence guarantees (vs. deterministic guarantees) and their convergence rate can be doubly-exponential in the size of the tree [19, 23] instead of at most exponential for minimax algorithms. Furthermore, while we believe extending our analysis to an MCTS-based algorithm is possible in principle, it presents a formidable technical challenge as the algorithm's states depend on visit counts and empirical rewards at each node, this would make the equations substantially more involved, and we view it as an important direction for future research. Finally,

although MCTS-based algorithms are state-of-the-art in multiple settings, exact minimax algorithms still remain practically useful in chess engines [24] and hybrid (RL+solving) approaches [22].

## 8   Discussion and limitations

**Discussion**   In this work, we introduced the *forward model*, a simple synthetic game model which provably addresses limitations of previous models while retaining some form of analytical tractability. For binary-valued trees, we established a closed-form expression for SOLVE's average-complexity and branching factor, and showed that the *F-games* were of adjustable difficulty and could span from maximally easy to maximally hard instances. For discrete-valued trees, we characterized the behavior of the TEST, ALPHA-BETA and SCOUT algorithms with equations, allowing a fast and efficient computation of the complexity and asymptotic branching factor, intractable with Monte-Carlo simulations. Unlike previous analysis under the *standard model*, we didn't manage to find closed-form expressions for the branching factors of all studied algorithms and we leave this open for future work. However, we established that TEST and ALPHA-BETA share the same branching factor, and we hypothesize that SCOUT shares it as well. This property was numerically confirmed by extensive numerical experiments, which further revealed that ALPHA-BETA incurred a larger sub-exponential factor than other approaches, suggesting that it is a poor baseline for practical scenarios.

**Limitations**   Our work focused on discrete-valued trees, which in our opinion best represent real-world games, and are numerically cheaper to solve. That being said, the equations we derived for the complexity analysis of TEST, ALPHA-BETA and SCOUT are, with small modifications, applicable to continuous values, opening avenues for an extended analysis of *F-games*. An important disclaimer is that we do not claim that our model accurately represents real-world games dynamics: it suffers from several important limitations that are consequences of its simplicity. First, the model only generates uniform $b$-ary trees of constant depth, which does not reflect the variable branching degrees and depths found in real games. Extending the analysis to more complex tree structures is a direction for future work. Second, the conditional distribution used to generate child values is the same for every node in the tree. It depends only on the parent's value, not on the broader game state. Furthermore, the method of ensuring the minimax constraint (truncating the distribution) is just one of many possibilities; other, more complex transformations could be valid but may also be harder to analyze. Regarding the choice of algorithms, we focused on TEST, SCOUT and ALPHA-BETA, but an analysis of MTD(f) and PVS would be an insightful extension, though they may prove more difficult to conduct since they require modeling the distribution of the test certificate value (and not only its binary outcome). However, we believe that SCOUT is a good proxy for PVS, and that PVS optimizations should not change asymptotic properties. Similarly, we think TEST-BISECTION is good proxy for MTD(f) where instead of choosing the next threshold using the certificate value of the last iterate—as in MTD(f)—we select it with a simple bisection rule. Of course, this is only speculation, which is why we believe it's important that PVS and MTD(f) are properly analyzed in the future under the *forward model*.

## 9   Broader impact

While our work focuses on classical deterministic game-solving algorithms, we believe it contributes to ongoing discussions at the intersection of learning, planning, and decision-making. Search remains a core ingredient of modern AI systems: top-performing agents such as AlphaZero combine learning with planning, yet the theoretical understanding of deterministic search methods (e.g., AlphaBeta) remains limited. Our analysis introduces a simple average-case model that incorporates structural dependencies in search trees, bridging a gap in the theoretical understanding of deterministic solvers. By providing a clearer picture of their expected behavior, our work may help guide the design of hybrid systems that integrate learning and deterministic planning.

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

# A   Proof material

In this section we provide detailed proofs for the results established in the paper, that were not included in the main text due to space and readability constraints.

## A.1   Proof of Theorem 4.1

We first recall the result:

**Theorem 4.1.** *For $b \in \mathbb{N}$ and for all $q \in [0, \frac{1}{b}]$, the branching factor of SOLVE satisfies $r_{SOLVE} = \mathcal{O}(b)$.*

*Proof.* Recall that

$$t(q, b) = \sum_{k=1}^{b-1} \frac{1 + (b - k - 1)q}{b} k(1 - q)^k.$$

By analyzing the sign of $\frac{\partial t}{\partial q}(q, b)$ and showing it's non-positive for $q \in [0, 1]$ we can establish that $t$ is monotonically decreasing in $q$ on $[0, 1]$. Then, we only have to show that $t_b = t(q = 1/b, b) = \mathcal{O}(b)$, for the result to be true for all $q \in [0, \frac{1}{b}]$. First we separate positive and negative sums:

$$t_b = \sum_{k=1}^{b-1} \frac{1 + (b - k - 1)\frac{1}{b}}{b} k(1 - \frac{1}{b})^k \tag{13}$$

$$= \frac{2}{b} \sum_{k=1}^{b-1} k(1 - \frac{1}{b})^k - \frac{1}{b^2} \sum_{k=1}^{b-1} k(k + 1)(1 - \frac{1}{b})^k. \tag{14}$$

By expanding both sums (with geometrical sum expansions) and using the fact that:

$$(1 - \frac{1}{b})^b = e^{-1} + o(1), \tag{15}$$

we can show that:

$$\frac{2}{b} \sum_{k=1}^{b-1} k(1 - \frac{1}{b})^k = (2 - 4e^{-1})b + o(b) \tag{16}$$

$$\text{and } \frac{1}{b^2} \sum_{k=1}^{b-1} k(k + 1)(1 - \frac{1}{b})^k = (2 - 5e^{-1})b + o(b), \tag{17}$$

allowing us to conclude that $t_b = e^{-1}b + o(b) = \mathcal{O}(b)$. This terminates the proof. As a side note, if we consider instead $q = \frac{1}{b^a}$ for $a > 1$, we can show that $t_b = \frac{b}{2} + o(b)$, which is slightly higher than $e^{-1}b$ (because $e^{-1} \approx 0.37 < \frac{1}{2}$) and asymptotically reaches the same constant as in Equation 4.   $\square$

## A.2   Proof of Theorem 5.1

We start by proving an intermediate property, useful for proving Theorem 5.1.

**Proposition A.1.** *Evaluating ALPHA-BETA with a given window is always more efficient than splitting this window into two sub-windows and evaluating these sub-windows separately.*

*For $h \geq 0$ and $x \in [-n, n]$, for all $\alpha, \beta \in \mathbb{N}$ such that $\alpha < \beta - 1$ and for all $\gamma \in \mathbb{N}$ such that $\alpha < \gamma < \beta$:*

$$I_{AB}^{x,\alpha,\beta}(h) \leq I_{AB}^{x,\alpha,\gamma}(h) + I_{AB}^{x,\gamma,\beta}(h).$$

*Proof.* Let's recall the expression of the complexity of ALPHA-BETA $I_{AB}^{x,\alpha,\beta}(h, c)$ and it's auxiliary function $J_{AB}^{x,\alpha,\beta}(h, c)$:

$$I_{AB}^{x,\alpha,\beta}(h, c) = \frac{1}{c} \left[ I_{AB}^{-x,-\beta,-\alpha}(h - 1, b) + \mathbb{1}_{\{x < \beta\}} J_{AB}^{x,\max(\alpha,x),\beta}(h, c - 1) \right]$$

$$+ \frac{c - 1}{c} \mathop{\mathbb{E}}_{\substack{X' \sim \mu \\ X' \geq -x}} \left[ I_{AB}^{X',-\beta,-\alpha}(h - 1, b) + \mathbb{1}_{\{-X' < \beta\}} I_{AB}^{x,\max(\alpha,-X'),\beta}(h, c - 1) \right], \tag{18}$$

$$J_{\text{AB}}^{x,\alpha,\beta}(h,c) = \mathop{\mathbb{E}}_{\substack{X'\sim\mu \\ X'\geq-x}} \left[ I_{\text{AB}}^{X',-\beta,-\alpha}(h-1,b) + \mathbb{1}_{\{-X'<\beta\}} J_{\text{AB}}^{x,\max(\alpha,-X'),\beta}(h,c-1) \right]. \quad (19)$$

We conduct the proof for the function $I$ and $J$ altogether, *i.e.* we want to prove that for all $\alpha < \gamma < \beta$ and all integers $h, c$ we have:

$$I_{\text{AB}}^{x,\alpha,\beta}(h,c) \leq I_{\text{AB}}^{x,\alpha,\gamma}(h,c) + I_{\text{AB}}^{x,\gamma,\beta}(h,c)$$

and

$$J_{\text{AB}}^{x,\alpha,\beta}(h,c) \leq J_{\text{AB}}^{x,\alpha,\gamma}(h,c) + J_{\text{AB}}^{x,\gamma,\beta}(h,c)$$

.

Let's proceed by double induction on the integers $h$ and $c$.

**Base cases** for all $h$, $I_{\text{AB}}^{x,\alpha,\beta}(h,0) = 0$ and $I_{\text{AB}}^{x,\alpha,\gamma}(h,0) + I_{\text{AB}}^{x,\gamma,\beta}(h,0) = 0$. Similarly for all $c > 0$, $I_{\text{AB}}^{x,\alpha,\beta}(0,c) = 1$ and $I_{\text{AB}}^{x,\alpha,\gamma}(0,c) + I_{\text{AB}}^{x,\gamma,\beta}(0,c) = 2$. It follows identically for J. So the base cases hold.

**Induction step** now we assume that the property holds for $(h, c-1)$ and $(h-1, b)$. We'll detail here the induction step for the function $J$, as it is less cumbersome to write, but the proof for $I$ follows the exact same steps. First, let's remark that the expectation involves a sum of terms, and let's try to prove the inequality holds term by term. Let $X' \geq -x$, first if $\mu(X') = 0$, the inequality holds trivially, so we consider without loss of generality $\mu(X') > 0$. Let's define:

$$A = I_{\text{AB}}^{X',-\beta,-\alpha}(h-1,b) + \mathbb{1}_{\{-X'<\beta\}} J_{\text{AB}}^{x,\max(\alpha,-X'),\beta}(h,c-1)$$

and

$$B = I_{\text{AB}}^{X',-\gamma,-\alpha}(h-1,b) + \mathbb{1}_{\{-X'<\gamma\}} J_{\text{AB}}^{x,\max(\alpha,-X'),\gamma}(h,c-1)+$$
$$I_{\text{AB}}^{X',-\beta,-\gamma}(h-1,b) + \mathbb{1}_{\{-X'<\beta\}} J_{\text{AB}}^{x,\max(\gamma,-X'),\beta}(h,c-1)$$

and show that $A \leq B$.

**Case 1** If $\beta < -X'$, then $\mathbb{1}_{\{-X'<\beta\}} = \mathbb{1}_{\{-X'<\gamma\}} = 0$, the property then holds using the induction hypothesis for $(h-1, b)$.

**Case 2** Now, if $\gamma \leq -X' < \beta$, $\mathbb{1}_{\{-X'<\beta\}} = 1$ and $\mathbb{1}_{\{-X'<\gamma\}} = 0$. Moreover, $\max(\gamma, -X') = -X'$ and $\max(\alpha, -X') = -X'$. So:

$$A = I_{\text{AB}}^{X',-\beta,-\alpha}(h-1,b) + J_{\text{AB}}^{x,-X',\beta}(h,c-1)$$

and

$$B = I_{\text{AB}}^{X',-\gamma,-\alpha}(h-1,b) + I_{\text{AB}}^{X',-\beta,-\gamma}(h-1,b) + J_{\text{AB}}^{x,-X',\beta}(h,c-1).$$

The inequality also directly holds using the induction hypothesis for $(h-1, b)$.

**Case 3** Now, if $\alpha \leq -X' < \gamma$, $\mathbb{1}_{\{-X'<\beta\}} = 1$ and $\mathbb{1}_{\{-X'<\gamma\}} = 1$. Moreover, $\max(\gamma, -X') = \gamma$ and $\max(\alpha, -X') = -X'$. So:

$$A = I_{\text{AB}}^{X',-\beta,-\alpha}(h-1,b) + J_{\text{AB}}^{x,-X',\beta}(h,c-1)$$

and

$$B = I_{\text{AB}}^{X',-\gamma,-\alpha}(h-1,b) + J_{\text{AB}}^{x,-X',\beta}(h,c-1) + I_{\text{AB}}^{X',-\beta,-\gamma}(h-1,b) + J_{\text{AB}}^{x,\gamma,\beta}(h,c-1).$$

Here again, the inequality holds by induction hypothesis on $(h, b)$ and using the fact $J_{\text{AB}}^{x,\gamma,\beta}(h, c-1)$ is non-negative.

**Case 4** Finally, if $-X' < \alpha$, we obtain:

$$A = I_{\text{AB}}^{X',-\beta,-\alpha}(h-1,b) + J_{\text{AB}}^{x,\alpha,\beta}(h,c-1)$$

and

$$B = I_{\text{AB}}^{X',-\gamma,-\alpha}(h-1,b) + J_{\text{AB}}^{x,\alpha,\beta}(h,c-1) + I_{\text{AB}}^{X',-\beta,-\gamma}(h-1,b) + J_{\text{AB}}^{x,\gamma,\beta}(h,c-1).$$

Here the inequality holds using both the induction steps at $(h-1, b)$ and $(h, c-1)$. We have covered all possible values of $X'$, so the proof is concluded. $\qquad\square$

Now we recall the theorem of interest:

**Theorem 5.1.** ALPHA-BETA *called with a full window* $\{-n, ..., n\}$ *is more efficient than the approach consisting in testing every possible value with the* TEST *procedure (i.e.,* TEST-BRUTEFORCE*) and* ALPHA-BETA *and* TEST *share the same asymptotic branching factor, in the precise sense that for* $h \geq 0$ *and* $x \in \{-n, ..., n\}$:

$$I_{\text{AB}}^{x,-n,n}(h) \leq \sum_{s=-n+1}^{n} I_{\text{TEST}}^{x,s}(h) \quad and \quad r_{\text{AB}} = r_{\text{TEST}}.$$

*Proof.* The first part of the theorem is obtained by iteratively applying Proposition A.1 with $\alpha = -n$ and $\beta = n$ and choosing $\gamma = -n + 1$, then $\gamma = -n + 2$ and so on, until $\gamma = n - 1$.
The second part can be deducted by remarking that we have:

$$\max_{s} I_{\text{TEST}}^{x,s}(h) \leq I_{\text{AB}}^{x,-n,n}(h) \leq \sum_{s} I_{\text{TEST}}^{x,s}(h) \leq 2n \max_{s} I_{\text{TEST}}^{x,s}(h).$$

The left-hand-side inequality reflects the fact that a smaller $\alpha{-}\beta$ window results in evaluating strictly less nodes — note that this doesn't hold for SCOUT, due to non-monotonicity of TEST's complexity with respect to the threshold value $s$. By taking power $1/h$ on both sides and taking the limit in $+\infty$ this gives us the desired result. $\square$

## A.3  Proof of Theorem 5.2

We first recall the result:

**Theorem 5.2.** SCOUT *called with a full window* $\{-n, ..., n\}$ *is more efficient than the approach consisting in testing every possible value with the* TEST *procedure (i.e.,* TEST-BRUTEFORCE*). For* $h \geq 0$ *and* $x \in \{-n, ..., n\}$:

$$I_{SCOUT}^{x,-n,n}(h) \leq \sum_{s=-n+1}^{n} I_{\text{TEST}}^{x,s}(h) \quad and \quad r_{SCOUT} \leq r_{\text{TEST}}.$$

*Proof.* The proof is very similar to that of Proposition A.1, in particular it relies on the same type of induction, so we only detail here the induction step for the function $J_{\text{SCOUT}}$.

We assume that

$$I_{\text{SCOUT}}^{x,-n,n}(h) \leq \sum_{s=-n+1}^{n} I_{\text{TEST}}^{x,s}(h)$$

and

$$J_{\text{SCOUT}}^{x,-n,n}(h) \leq \sum_{s=-n+1}^{n} J_{\text{TEST}}^{x,s}(h).$$

We recall the expression for $J_{\text{SCOUT}}$:

$$J_{\text{SCOUT}(h,c)}^{x,\alpha,\beta} = \mathbb{E}_{X' \geq -x} \left[ I_{\text{TEST}(h-1,b)}^{X',-\alpha} + \mathbb{1}_{\{\alpha < -X'\}} I_{\text{SCOUT}(h-1,b)}^{X',-\beta,-\alpha-1} + \mathbb{1}_{\{-X'<\beta\}} J_{\text{SCOUT}(h,c-1)}^{x,\max(\alpha,-X'),\beta} \right]. \quad (20)$$

A little subtlety that was not explicit in Equations 11 and 12 is that for the special case $\alpha = \beta$, Algorithm 4 terminates instantly, incurring a cost of 0. As a consequence when called with $\alpha = s - 1$ and $\beta = s$, SCOUT is equivalent to a call to TEST($s$), since $I_{\text{SCOUT}(h-1,b)}^{X',-\beta,-\alpha-1} = I_{\text{SCOUT}(h-1,b)}^{X',-s,-s} = 0$. Without loss of generality, we consider in the following $\alpha < \beta - 1$, as the desired inequality is clearly true for $\alpha = \beta - 1$. Let's show the inequality holds term by term for every value of $X'$. We define:

$$A = I_{\text{TEST}(h-1,b)}^{X',-\alpha} + \mathbb{1}_{\{\alpha<-X'\}} I_{\text{SCOUT}(h-1,b)}^{X',-\beta,-\alpha-1} + \mathbb{1}_{\{-X'<\beta\}} J_{\text{SCOUT}(h,c-1)}^{x,\max(\alpha,-X'),\beta}$$

and

$$B = \sum_{s=\alpha+1}^{\beta} I_{\text{TEST}(h-1,b)}^{X',-s+1} + \sum_{s=\alpha+1}^{\beta} \mathbb{1}_{\{-X'<s\}} J_{\text{TEST}(h,c-1)}^{x,s}.$$

Let's show that $A \leq B$ in all cases.

**Case 1** If $\beta \leq -X'$, then $\mathbb{1}_{\{\alpha < -X'\}} = 1$ and $\mathbb{1}_{\{-X' < \beta\}} = 0$. So:

$$A = I_{\text{TEST}(h-1,b)}^{X',-\alpha} + I_{\text{SCOUT}(h-1,b)}^{X',-\beta,-\alpha-1}$$

and

$$B = \sum_{s=\alpha+1}^{\beta} I_{\text{TEST}(h-1,b)}^{X',-s+1} + \sum_{s=\alpha+1}^{\beta} \mathbb{1}_{\{-X' < s\}} J_{\text{TEST}(h,c-1)}^{x,s} \geq \sum_{s=\alpha+1}^{\beta} I_{\text{TEST}(h-1,b)}^{X',-s+1}.$$

By induction hypothesis on $(h-1,b)$ and variable change $s' = -s+1$, we can write:

$$A \leq I_{\text{TEST}(h-1,b)}^{X',-\alpha} + \sum_{s=-\beta+1}^{-\alpha-1} I_{\text{TEST}(h-1,b)}^{X',s} \leq I_{\text{TEST}(h-1,b)}^{X',-\alpha} + \sum_{s=\alpha+2}^{\beta} I_{\text{TEST}(h-1,b)}^{X',-s+1} \leq \sum_{s=\alpha+1}^{\beta} I_{\text{TEST}(h-1,b)}^{X',-s+1} \leq B.$$

So the inequality holds in this case.

**Case 2** If $\alpha < -X' < \beta$, then $\mathbb{1}_{\{\alpha < -X'\}} = 1$ and $\mathbb{1}_{\{-X' < \beta\}} = 1$ and $\max(\alpha, -X') = -X'$. $A$ and $B$ become:

$$A = I_{\text{TEST}(h-1,b)}^{X',-\alpha} + I_{\text{SCOUT}(h-1,b)}^{X',-\beta,-\alpha-1} + J_{\text{SCOUT}(h,c-1)}^{x,-X',\beta}$$

and

$$B = \sum_{s=\alpha+1}^{\beta} I_{\text{TEST}(h-1,b)}^{X',-s+1} + \sum_{s=\alpha+1}^{\beta} \mathbb{1}_{\{-X' < s\}} J_{\text{TEST}(h,c-1)}^{x,s}.$$

For the term in $(h-1,b)$ it's the same as in Case 1. For the terms in $(h,c-1)$, we remark that

$$\sum_{s=\alpha+1}^{\beta} \mathbb{1}_{\{-X' < s\}} J_{\text{TEST}(h,c-1)}^{x,s} = \sum_{s=-X'+1}^{\beta} J_{\text{TEST}(h,c-1)}^{x,s}$$

By induction hypothesis on $(h,c-1)$, with $\alpha = -X'$, we have:

$$J_{\text{SCOUT}(h,c-1)}^{x,-X',\beta} \leq \sum_{s=-X'+1}^{\beta} J_{\text{TEST}(h,c-1)}^{x,s}.$$

So the inequality holds term by term for this case.

**Case 3** If $-X' \leq \alpha$, we have then:

$$A = I_{\text{TEST}(h-1,b)}^{X',-\alpha} + J_{\text{SCOUT}(h,c-1)}^{x,-X',\beta}$$

and

$$B = \sum_{s=\alpha+1}^{\beta} I_{\text{TEST}(h-1,b)}^{X',-s+1} + \sum_{s=\alpha+1}^{\beta} \mathbb{1}_{\{-X' < s\}} J_{\text{TEST}(h,c-1)}^{x,s}.$$

By reusing arguments from the two previous cases, we can see easily that this case holds as well, thus concluding the proof. $\square$

# B  Monte-Carlo Simulations

In this section we provide results of Monte-Carlo simulations, experimentally validating the equations characterizing the different algorithms in the paper. All experiments here, and in the main text, were run in a couple of hours of CPU time on a consumer-grade laptop.

In Figure 3, 4 and 5, we represent the evolution of the Monte-Carlo mean estimator of the TEST, SCOUT and ALPHA-BETA complexities, respectively. The Monte-Carlo estimator is represented as a function of the number of trials, for different settings of distribution $\mu$, branching degree $b$, value range $n$ and tree height $h$. In every scenario, the Monte-Carlo estimator converges to the oracle computed using equations derived in Section 5. The settings were chosen to showcase a high diversity of parameters, while maintaining the computational cost reasonable. Results are averaged over 5 independent random seeds. Shaded areas represent bootstrapped 95% confidence interval.

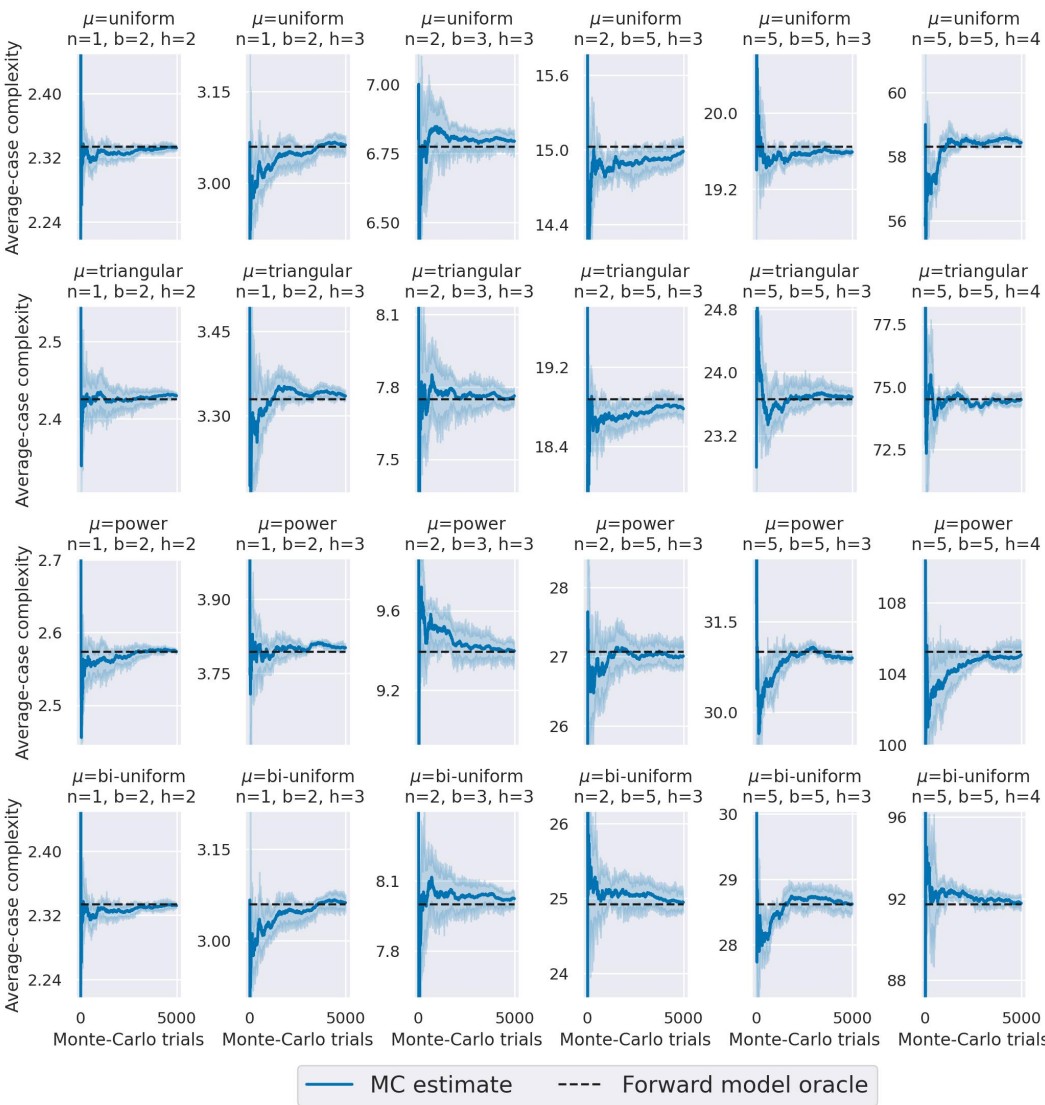

Figure 3: Evolution of the Monte-Carlo mean estimator of the TEST complexity, as a function of the number of trials, for different settings. Results are averaged over 5 independent random seeds and shaded areas represent bootstrapped 95% confidence interval. The oracle is computed using Equations 7 and 8.

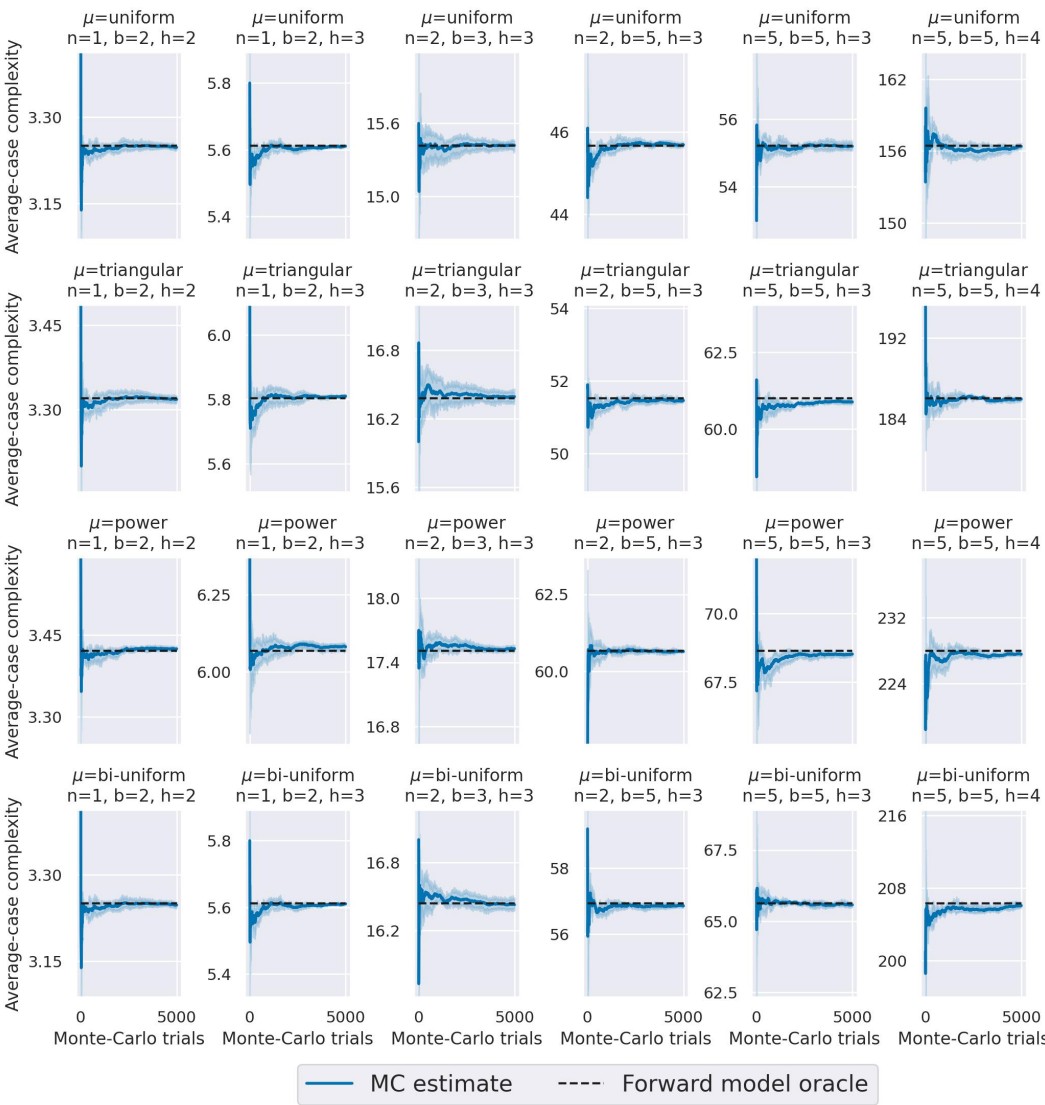

Figure 4: Evolution of the Monte-Carlo mean estimator of the ALPHA-BETA complexity, as a function of the number of trials, for different settings. Results are averaged over 5 independent random seeds and shaded areas represent bootstrapped 95% confidence interval. The oracle is computed using Equations 9 and 10.

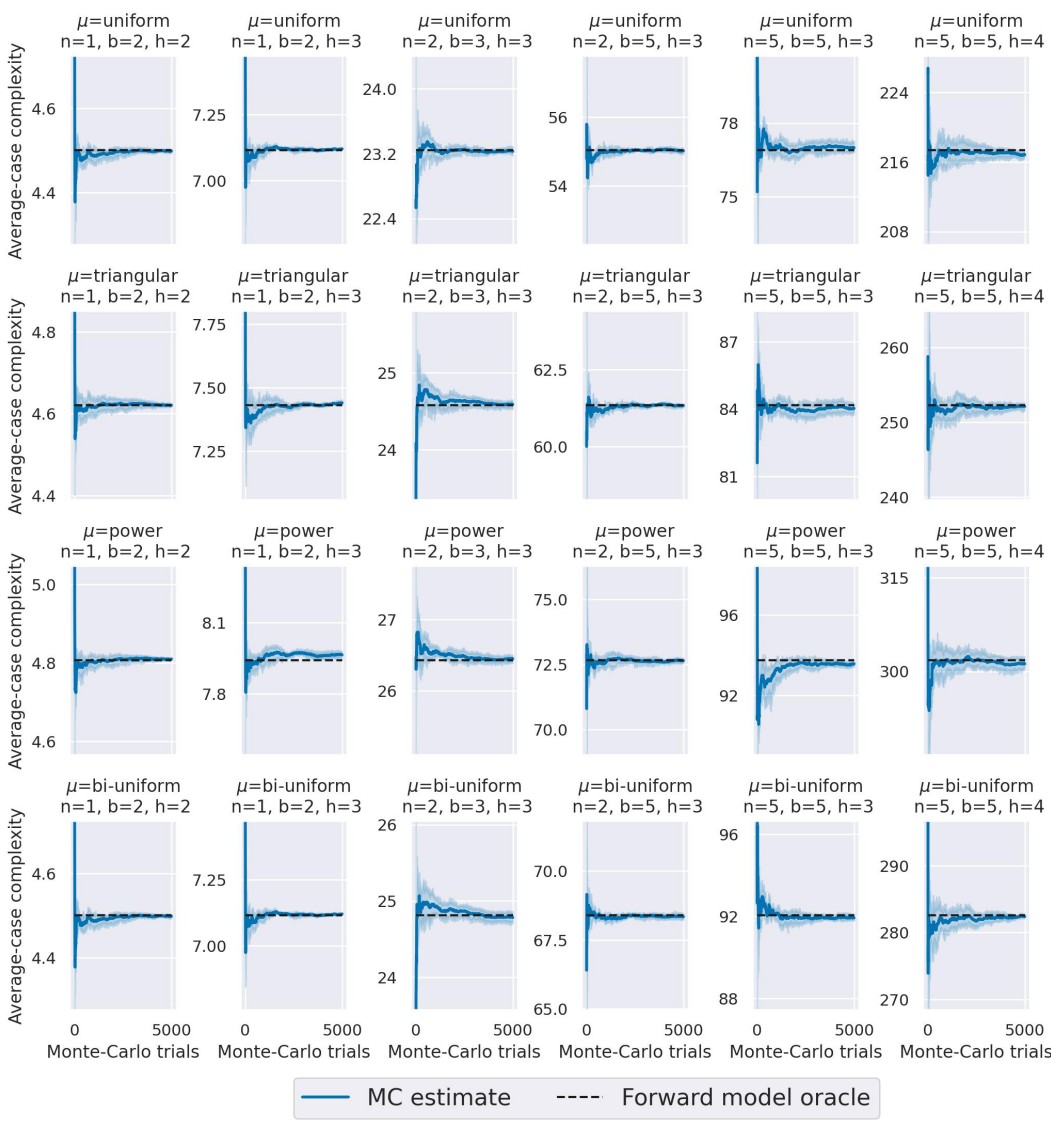

Figure 5: Evolution of the Monte-Carlo mean estimator of the scout complexity, as a function of the number of trials, for different settings. Results are averaged over 5 independent random seeds and shaded areas represent bootstrapped 95% confidence interval. The oracle is computed using Equations 11 and 12.

## C Derivation details for the SOLVE analysis

---

**Algorithm 5:** SOLVE($N, h$) — binary-valued tree

---

**Input:** Current node $N$, search depth $h$
**Output:** Value of node $N$.
**if** $h = 0$ **then return** *N.value*
$best \leftarrow 0$
**foreach** *N' in N.children* **do**
  $\quad value \leftarrow 1 - \text{SOLVE}(N', h - 1)$
  $\quad best \leftarrow \max(best, value)$
  $\quad$ **if** $best = 1$ **then break**
**end**
**return** $best$

---

In the following, we provide additional details for the analysis of the SOLVE algorithm.

We analyze the average-case complexity $I_{\text{SOLVE}}(h)$ of SOLVE on a depth-$h$ tree generated by the *forward model* where $\mu = \mathcal{B}(q)$ (Bernoulli distribution with $q$ the probability of drawing a 0). Clearly, $I_{\text{SOLVE}}(h) = \mathbb{E}_{X \sim \mu} I_{\text{SOLVE}}^X(h)$ where $I_{\text{SOLVE}}^x(h)$ denotes the complexity of SOLVE, but conditioned on the evaluated node value $x \in \{0, 1\}$. By capturing in equations the execution flow described in Algorithm 5 for every encountered case, we can characterize the dynamics of SOLVE. If $x = 0$ all children values $x_i'$s are 1. In this case SOLVE will recursively evaluate all $b$ children:

$$I_{\text{SOLVE}}^0(h) = bI_{\text{SOLVE}}^1(h - 1). \tag{21}$$

Now, if $x = 1$, at least one child $x_i'$ will hold the value 0, and whenever SOLVE finds it, it will terminate early. Hence, SOLVE will incur the cost of evaluating this $x_i' = 0$ child plus the expected number of failed trials needed to find it, multiplied by the cost of evaluating a $x_i' = 1$ child:

$$I_{\text{SOLVE}}^1(h) = I_{\text{SOLVE}}^0(h - 1) + t(q, b)I_{\text{SOLVE}}^1(h - 1), \tag{22}$$

where $t(q, b)$ is the expected number of trials before finding a child with value 0. Note that we can write these equations because by design, the distribution $\mu$ is independent from the height $h$ (given the knowledge of the value $x$). Under the *forward model*, $t$ can be derived by compounding the individual probabilities of finding a child with value 0 on the first $(b - 1)$ trials, leading to the expression in Equation 5:

$$t(q, b) = \sum_{k=1}^{b-1} \frac{1 + (b - k - 1)q}{b} k(1 - q)^k.$$

Equations 21 and 22 together define a recursive linear system, with initial conditions $I_{\text{SOLVE}}^x(0) = 1$ (a tree with only one node always incurs a cost of 1). Luckily, a closed-form solution for $I_{\text{SOLVE}}(h)$ can be derived, first we define:

$$r_{1,2} = \frac{t(q, b) \pm \sqrt{t(q, b)^2 + 4b}}{2}, \tag{23}$$

allowing us to write:

$$I_{\text{SOLVE}}(h) = q\left(Ar_1^h + (1 - A)r_2^h\right) + b(1 - q)\left(Ar_1^{h-1} + (1 - A)r_2^{h-1}\right) \tag{24}$$

where $A \in [0, 1]$ is defined as:

$$A = \frac{1}{2} + \frac{1 + t(q, b)/2}{\sqrt{t(q, b)^2 + 4b}}.$$

Clearly the branching factor is determined by the larger of $r_1$ and $r_2$, hence the expression in Equation 5:

$$r_{\text{SOLVE}} = r_1 = \frac{t(q, b) + \sqrt{t(q, b)^2 + 4b}}{2}.$$

