# OpenReview forum: "AlphaBeta is not as good as you think: a simple class of synthetic games for a better analysis of deterministic game-solving algorithms"
_NeurIPS.cc/2025/Conference — NeurIPS 2025 poster_

### Official Review · Reviewer_cYDt · 2025-06-20

**Clarity:** 4
**Significance:** 2
**Originality:** 3
**Rating:** 4
**Confidence:** 3

**Summary:**

The authors propose a new probabilistic model for constructing game trees to evaluate deterministic solvers. In the standard model, leaf values are independently sampled from a fixed distribution and propagated up the tree according to minimax constraints. The authors argue that this strips the games of structural complexity and results in less meaningful analysis. The proposed forward model instead samples a game value at the root and recursively samples child values from a conditional distribution on the parent's value, while still ensuring minimax constraints are respected. This leads to more realistic and harder games for average-case analysis. This analysis leads to the interesting result that AlphaBeta search incurs a large multiplicative constant on its expected branching factor, which seems to have significant practical performance implications.

**Questions:**

1. What are “realistic” parameterizations for the forward model?
2. In light of 1, can the authors please justify their parameterizations?
3. Are there $\mu$ where AlphaBeta outperforms the other algorithms?
3. Can the authors elaborate on the modifications or additional assumptions needed for the continuous case?

**Ethical Concerns:**

["NO or VERY MINOR ethics concerns only"]

**Final Justification:**

Upon reading the final reviews and discussion, I would be fine with the paper being accepted. Concerns with broader impact still stand, so I will maintain my borderline rating.

**Limitations:**

The authors discuss certain limitations about the work related to the analysis of the forward model and the discrete setting. However, as mentioned in the Weaknesses section, the authors do not discuss the potential limitations of the forward model itself.

**Paper Formatting Concerns:**

I did not notice any formatting concerns.

**Quality:**

3

**Strengths And Weaknesses:**

**Strengths**
* The paper is well-written and generally very easy to follow.
* The forward model is simple to understand, implement, and seems to offer convenient analysis for very large trees. I see it as a significant contribution to the area.
* The numerical analysis offers a hypothesis which validates long-standing observations about the practical advantages to using Scout over AlphaBeta.
* Though I have some questions/comments about limitations and I’m not completely familiar with all of the cited literature, I think the authors have somewhat accomplished the title goal of providing a new model that enables better analysis of deterministic game-solving algorithms.

**Weaknesses**
* The major weakness I feel this works has is its lack of discussion related to the limitations of the forward model. Similar to how the standard model does not enforce “ancestor dependency”, are there important structural game tree properties not present under the forward model? I feel some discussion on this could be added to the paper.
* One area where this shows up is the experiments in Section 6. The parameterization of the forward model starts with, in my opinion, a relatively unrealistic set of distributions. Fairly or not, this may have a significant effect on AlphaBeta, because the distribution choice may influence how frequently and rapidly AlphaBeta degenerates to the null window. I think this is worth investigating further as I noticed that AlphaBeta seems to suffer less in the bimodal-uniform case.
* The paper would be stronger if the analysis was extended to include a real game, even just to justify parameterization choices for the forward model.
* Though this work seems to make progress toward answering important questions about deterministic solvers, it is worth mentioning that its applicability to the broader community at NeurIPS may be somewhat limited.

---

> ### Author Rebuttal · Authors · 2025-07-31
>
> We thank the reviewer for the thoughtful and constructive feedback. We are particularly grateful for the recognition of our model’s simplicity, analytical convenience, and its implications for practical algorithm evaluation.
>
> **On the limitations of the work**
>
> >“The major weakness I feel this works has is its lack of discussion related to the limitations of the forward model. Similar to how the standard model does not enforce “ancestor dependency”, are there important structural game tree properties not present under the forward model? I feel some discussion on this could be added to the paper.”
>
> This is a fair point. We agree that our initial submission focused more on the limitations of our **analysis** than on the limitations of the **forward model itself**. We have added a paragraph to the manuscript to address this, and we will summarize the key limitations here.
>
> The forward model was intentionally designed to be **simple**, as this is a crucial requirement to enable a tractable mathematical analysis. This simplicity naturally imposes some limitations:
>
> 1. **Fixed Tree Structure:** The model only generates uniform $b$-ary trees of constant depth, which does not reflect the variable branching factors and depths found in real games. Extending the analysis to more complex tree structures is a direction for future work.
> 2. **Simplified Dependencies:** The conditional distribution used to generate child values is the same for every node in the tree. It depends only on the parent's value, not on the broader game state. Furthermore, the method of ensuring the minimax constraint (truncating the distribution) is just one of many possibilities; other, more complex transformations could be valid but may also be harder to analyze.
>
> The goal of the forward model is not to perfectly replicate the complexity of real-world games, but to capture a crucial property (ancestor dependency) missing from the standard model, while remaining analyzable.
>
> **On the parametrization of the forward model.**
> >“The parameterization of the forward model starts with, in my opinion, a relatively unrealistic set of distributions. Fairly or not, this may have a significant effect on AlphaBeta, because the distribution choice may influence how frequently and rapidly AlphaBeta degenerates to the null window. I think this is worth investigating further as I noticed that AlphaBeta seems to suffer less in the bimodal-uniform case.”
>
>
> >1. What are “realistic” parameterizations for the forward model?
> >2. In light of 1, can the authors please justify their parameterizations?
> >3. Are there mu where AlphaBeta outperforms the other algorithms?
>
> These questions are linked, so we will address them together.
>
> First, to clarify our experimental findings: across all distributions we tested (including many randomly generated ones not shown in the paper), we did not find any where the fundamental conclusion changed and AlphaBeta outperformed Scout.
>
> Our choice of distributions for the figures was deliberate, aiming to test the algorithms under diverse and informative conditions:
> - The **uniform distribution** is a natural starting point, representing the case with the maximum possible diversity (entropy) in game values.
> - The **triangular, power and bi-modal distributions** were chosen to create problems of **increasing theoretical difficulty**, pushing the algorithms' performance closer to their asymptotic worst-case. The observation that AlphaBeta suffers less here is interesting, but it still does not outperform Scout.
>
> Regarding "realistic" parameterizations, it is an open question whether any parameterization of our simple model could accurately capture a specific real-world game. As discussed above, the model is likely **too simple for that** (and this should have been made clearer in the paper). However, its strength lies in its ability to generate a continuous spectrum of game difficulties. We can tune its parameters to create games that are maximally easy, maximally hard (from a theoretical standpoint), and everything in between. This makes it a useful tool for stress-testing algorithms in a controlled manner.
>
>
>
> **On the applicability of our work to Neurips**
> >"Though this work seems to make progress toward answering important questions about deterministic solvers, it is worth mentioning that its applicability to the broader community at NeurIPS may be somewhat limited."
>
> The reviewer is right in pointing out the fact that the broader applicability of our work to NeurIPS may seem limited to some readers. To clarify that, we added a *Broader Impact* statement to the paper.
>
> While our work is centered on classical deterministic game-solving algorithms, we believe it is **timely and relevant** for the NeurIPS community for the following reasons.
>
> Search remains a key component of today’s top-performing agents in RL and game-solving, e.g. AlphaZero, MuZero, and others **combine learning with planning**. Even though MCTS is dominant, we believe deterministic methods like AlphaBeta still have value (see [1] for instance), especially when lower variance or stronger guarantees are needed.
>
> Our main contribution is a new average-case analysis model that, we hope, fills a **longstanding gap**. Despite decades of use, there hasn’t been much rigorous analysis of how these algorithms behave on more realistic trees. Our forward model brings in some structure (like ancestor dependencies) while staying simple enough to analyze. Hopefully, this helps uncover practical differences between algorithms that aren’t visible in the standard model.
>
> We believe this kind of theoretical insight could inform future hybrid systems that mix learning with deterministic planning. It also connects with broader NeurIPS themes around **decision-making**, **optimization**, and understanding the foundations of **intelligent behavior**.
>
>
> **On the Continuous Case**
> >“Can the authors elaborate on the modifications or additional assumptions needed for the continuous case?”
>
> We believe that no significant additional assumptions are needed to extend the analysis to the continuous case, though the analysis itself becomes more complex. The recursive equations are conceptually the same, with the main difference being that expectations become **integrals** over continuous variables rather than discrete sums.
>
> This has two main consequences:
> - The boundary conditions in the equations simplify slightly (e.g., in the discrete case $x > -s$ is equivalent to $x >= -s+1$, whereas in the continuous case they are the same event regardless of the inequality being strict or not, up to a set of measure zero).
> - The analysis becomes more difficult because we now have a system of **integral equations**. Solving these may require finding a closed-form solution or using numerical approximation methods (like discretization), which would effectively return us to a scenario similar to the discrete case.
>
> In summary, the framework extends directly, but the mathematical machinery required is more challenging.
>
> **References**
>
> [1] Cohen-Solal, Q., & Cazenave, T. (2023, May). Minimax Strikes Back. In Proceedings of the 2023 International Conference on Autonomous Agents and Multiagent Systems (pp. 1923-1931).

---

> > ### Comment · Reviewer_cYDt · 2025-08-05
> >
> > Thank you to the authors for their response. I have read all the reviews and the authors' responses.
> >
> > The authors have addressed several of my questions, and I believe the proposed additional limitation discussion will improve the paper's framing. Though I still have some reservations about the broader impact of the paper, the authors raise a good point concerning its potential connection to learning and planning.

---

### Official Review · Reviewer_zdYH · 2025-07-02

**Clarity:** 2
**Significance:** 2
**Originality:** 3
**Rating:** 4
**Confidence:** 2

**Summary:**

This paper introduces a forward model for analyzing deterministic game-solving algorithms, addressing the limitations of the standard model (e.g., independence assumption) by enforcing ancestor dependencies in game-tree construction. The authors derive recursive complexity equations for algorithms like AlphaBeta and Scout, showing that while all algorithms converge to the same asymptotic branching factor, finite-depth analysis reveals AlphaBeta incurs a larger constant factor, making it less efficient than Scout in practice. The model enables adjustable problem difficulty and rigorous comparisons via recursive formulas, validated by Monte Carlo simulations.

**Questions:**

Since this paper introduces a new forward model, which is claimed to be more "realistic", how does its tree distribution differ from that of standard random models?

**Ethical Concerns:**

["NO or VERY MINOR ethics concerns only"]

**Final Justification:**

The authors have addressed my concerns. And I raise my score to 4.

**Limitations:**

The authors have discussed the limitations.

**Paper Formatting Concerns:**

No.

**Quality:**

2

**Strengths And Weaknesses:**

**Strength**

- Theoretical Innovation: The paper introduces a novel forward model that addresses the long-criticized independence assumption in standard game-tree analysis. By enforcing ancestor dependencies, the authors claim that the model captures structural complexities of real-world games (e.g., chess, Go) more realistically, enabling adjustable difficulty calibration for algorithm evaluation.

- Rigorous Analysis: The recursive complexity equations for algorithms like AlphaBeta and Scout under the forward model provide a solid framework for theoretical comparison. The numerical validation via Monte Carlo simulations enhances the credibility of results.

**Weakness**

- The authors claim the forward model better represents real-world games by introducing ancestor dependencies, but fail to Provide concrete examples or case studies (e.g., in chess endgame search) to illustrate how ancestor dependencies mirror actual game dynamics. Thus, there no idea whether the introducing of the forward model is important.

- The literature review focuses on classic studies (mostly pre-2000) and the authors need to check if there are new works in this area. More discussion and comparison may need to be conducted.

---

> ### Author Rebuttal · Authors · 2025-07-31
>
> We thank the reviewer for their feedback and the opportunity to elaborate on our work's motivation and context.
>
> **On the Realism of the Forward Model**
> >“The authors claim the forward model better represents real-world games by introducing ancestor dependencies, but fail to Provide concrete examples or case studies (e.g., in chess endgame search) to illustrate how ancestor dependencies mirror actual game dynamics. Thus, there is no idea whether the introducing of the forward model is important.”
>
>
> This is a crucial point, and we appreciate the chance to clarify the role of our model. The importance of *ancestor dependencies* for modeling real-world games has been a long-standing topic in the literature (see [1] and [2] for instance). The standard model's i.i.d. assumption is widely acknowledged as its primary limitation.
>
> A classic example, as the reviewer suggests, comes from chess. If White is in a clearly winning position (e.g., up a queen), one expects that the vast majority of reachable terminal states in the subsequent subgame will be wins for White. The standard model fails to capture this strong **correlation** between a parent node's value and its children's values.
>
> Our forward model was designed as a **mathematically tractable** way to introduce exactly this type of correlation. While it does not pretend to generate real-world games per se, its purpose is to create a richer benchmark than the often-degenerate trees produced by the standard model, while preserving the ability to perform rigorous mathematical analysis (a feature that other proposals, like the incremental model, have struggled with).
>
> **On the literature review**
>
> >"The literature review focuses on classic studies (mostly pre-2000) and the authors need to check if there are new works in this area. More discussion and comparison may need to be conducted."
>
> We agree that a broader literature review would strengthen the paper. The reason we primarily cited older works is that the critiques we build upon were most clearly formulated in the earlier literature (e.g., Knuth & Moore, Pearl, Nau). In fact, our model can be seen as a response to issues that were already well-known at the time, and which inspired alternative approaches like the  *incremental model* which has proven hard to analyze formally.
>
> To our knowledge, no recent works have revisited this line of theoretical average-case analysis for deterministic game-solving algorithms. In contemporary game-playing research, the focus has shifted toward empirical algorithm design (e.g., MCTS, neural search policies), which tends to avoid formal models altogether. That said, if the reviewers are aware of relevant recent work that we may have missed, we would be grateful to include and discuss it in the final version.
>
> **On the difference in the tree distribution**
> >“Since this paper introduces a new forward model, which is claimed to be more "realistic", how does its tree distribution differ from that of standard random models?”
>
> The most significant difference lies in the distribution of the root value of the generated game trees. It is arguably logical to focus on the root-value distribution, as this is the output of a game-solving algorithm, and a robust testbed should include trees with diverse root values.
>
> Let's consider the binary-valued case for a clear comparison. Under the standard model, the root value distribution is highly constrained. For any leaf value distribution, as the tree depth grows, the root value distribution collapses to one of two outcomes: it is either deterministic (always 0 or always 1) or, at a single critical probability $q*$, it is a Bernoulli($q*$). The standard model is therefore unable to generate a wider spectrum of tree distributions.
>
> In contrast, our forward model is explicitly designed to generate a richer variety of trees. The root value distribution is directly parameterized and can be set to a Bernoulli($q$) for any $q$ between 0 and 1. As we demonstrate in Section 4, varying $q$ allows us to span the **full spectrum of game difficulty**, from the easiest case to the hardest. This shows that the forward model can generate a much larger diversity of game instances, making it an arguably richer model.
>
> **References**
>
> [1] Nau, D. S. (1982). An investigation of the causes of pathology in games. Artificial Intelligence, 19(3), 257-278.
>
> [2] Knuth, D. E., & Moore, R. W. (1975). An analysis of alpha-beta pruning. Artificial intelligence, 6(4), 293-326.

---

> > ### Comment · Reviewer_zdYH · 2025-08-05
> >
> > Thank you for your response.
> >
> > While the chess example illustrates a scenario where parent-child node value correlations exist, this alone does not sufficiently justify the necessity of introducing ancestor dependencies via the proposed forward model. Specifically, such correlations can often be naturally captured within the standard model when equilibrium strategies are employed.
> >
> > Can quantitative comparisons show that the proposed forward model better aligns with empirical game dynamics than the standard models?

---

> ### Author Response · Authors · 2025-08-06
>
> Thank you for engaging in the discussion with us. You are absolutely right saying that parent-children value correlation always exists in any game, and in particular in instances of the standard model. This is the case because the parent minimax value is a deterministic function of the children values. What we wanted to illustrate with the chess example (and we reckon it was poorly phrased in our rebuttal) is the fact that in real-world games there is typically **correlation among siblings,** e.g. two children of the same node are likely to hold similar values, this property is not modeled at all by the standard model. This exact point has been clearly articulated by Nau in 82’ (but also by Knuth in 75’). We include relevant quotes in the comment below due to character limits.
>
> > Can quantitative comparisons show that the proposed forward model
> > better aligns with empirical game dynamics than the standard models?
>
>
> This relates closely to our earlier discussion with **reviewer cYDt** on the limitations of our work. Our model is likely too simple to model a real-world game: they typically have a more complex tree structure, e.g. with early terminations; and the way children depend on parents is more varied than what we encode in the forward model (we use the same conditional distribution at each level, with a simple truncation method). This is the case because we wanted our model to retain some form of **mathematically tractability**.
>
>
> That being said, we can illustrate that a real-world game typically has correlation between siblings, i.e. ancestor dependencies as we originally meant it. These correlations are inherently impossible to model under i.i.d assumption.  For simplicity we choose a simpler version of Tic-Tac-Toe, modified such that no draw is possible: in case of a draw, the second player wins. This simplification allows us to set ourselves in a convenient binary-valued setting while keeping the same game dynamics.
>
> We want to measure **sibling correlation**: if we randomly pick two children of the same position, how correlated are their values? For this:
>
> -   We enumerate all positions after 3 moves (i.e., subgames where 6 actions remain), of which there are 252 (we chose this setting to obtain a large enough number of subgames).
>
> -   For each position, we randomly pick two children, compute their values, and store these values across all 252 positions.
>
> -   We repeat this 256 times (sampling different child pairs), and compute the average Pearson correlation between the two series of 252 values.
>
> -   The result is a sibling correlation of **0.109 ± 0.004** in Tic-Tac-Toe (mean$\pm$standard error).
>
>
> Now consider the **standard model**. We generate 32 random instances with 252 subtrees each (using varying values of the parameter _p_), and perform the same sibling-correlation analysis. Unsurprisingly, in all cases, the average correlation is near zero, regardless of $p$, because sibling values are independent by construction.
>
>
> And what about the **forward model**? We can repeat the exact same experiment. The numerical results show that depending on the parameter $q$, the forward model generates instances with increasing sibling correlations. See the table below.
>
>
> |Tic-Tac-Toe | Standard model | Forward model (q=0.25) | Forward model (q=0.5) | Forward model (q=0.75) |
> |--|--|--|--|--|
> |$0.109\pm0.004$  | $0.004\pm0.004$  | $0.076 \pm 0.004$ |$0.316\pm0.007$ | $0.631\pm0.010$|
>
>
> We do **not** claim that the forward model effectively models the Tic-Tac-Toe game, but at least, it can generate games with this kind of ancestor dependencies (sibling correlations), which is not the case of the standard model. As we show in the paper, this translates into the theoretical capacity of the forward model to generate richer distributions of root values and a wider range of game difficulty, including maximally hard instances.
>
> We thank you for raising this point. We agree that clarifying what we mean by "ancestor dependencies" will improve the paper’s presentation, and we hope this concrete example helps clarify both our motivation and our contributions. Please tell us if you have more questions.

---

> > ### Author Response · Authors · 2025-08-06
> > **Quotes from Nau and Knuth**
> >
> > Nau 1982a, sec 5 [1]:
> > > We now discuss another major difference between Pearl's game [the standard model] and games such as chess or checkers [...]. In games such as chess or checkers, the evaluation function value of a node is usually positively correlated with the evaluation function value of its parent. This also occurs in Pearl's game. However, there is another kind of dependency among node values which Pearl's game does not have. In games such as chess or checkers, positions are often characterized as 'strong' and 'weak'. Strong nodes are likely to be win nodes, and are likely to have high minimax values. Weak nodes are likely to be loss nodes, and are likely to have low minimax values. Since board positions change incrementally, a strong node is likely to have strong children, and a weak node is likely to have weak children. Thus the minimax values of sibling nodes (or other closely related nodes) are likely to be similar. Therefore, the game tree is likely to be differentiated into sections containing many strong nodes and few weak nodes, and sections containing many weak nodes and few strong nodes. The above property does not occur in Pearl's game. In particular, let P be a Pearl game, and let g and g' be any two nodes at the same depth in P. Then the minimax values of g and g', being functions of independent random variables, are independent of each other.
> >
> > Knuth 1975, sec 9 [2]:
> > >Our model [the standard model] gives independent values to all the terminal positions, but such independence does not happen very often in real games. For example, if f(p) [the value of the position] is based on the piece count in a chess game, all the positions following a blunder will tend to have low scores for the player who loses his men.
> >
> >
> > [1] Nau, D. S. (1982). An investigation of the causes of pathology in games. Artificial Intelligence, 19(3), 257-278.
> > [2] Knuth, D. E., & Moore, R. W. (1975). An analysis of alpha-beta pruning. Artificial intelligence, 6(4), 293-326.

---

### Official Review · Reviewer_8x4a · 2025-07-03

**Clarity:** 4
**Significance:** 3
**Originality:** 3
**Rating:** 5
**Confidence:** 3

**Summary:**

This paper proposes a new model for analyzing average-case complexity of determinsitic game-solving algorithms.
Previously, the "standard" model sampled leaf values to a tree of branching factor $b$ and depth $d$ from a fixed distribution.
The paper proposes a new "forward" model, where intermediate values are sampled progressively, level by level, until leaf nodes are reached.

The standard model is too simplistic, as shown by previous works, as the value asymptotically collapses to a single fixed value for all distributions. The new forward model allows for asymptotic analysis on deep trees much more efficiently than Monte-Carlo simulations.
Authors show in the asymptotic regime a result analogous to those of the standard model.
However, Alpha-Beta incurs a multiplicative constant proportional to the game’s value range, compared to other algorithms like Scout.

**Questions:**

Does the level dependence in the forward model increase probability of leaf-disjoint strategies (as defined in [1]), compared to to the standard model?

[1] Lorenz, U. and Monien, B. (2002). The secret of selective game tree search, when using random-error evaluations.
In STACS 2002: 19th Annual Symposium on Theoretical Aspects of Computer Science Antibes-Juan les Pins, France, March 14–16, 2002 Proceedings 19, pages 203–214. Springer.

**Ethical Concerns:**

["NO or VERY MINOR ethics concerns only"]

**Final Justification:**

All raised comments were resolved and I maintain score 5: accept.

**Limitations:**

yes

**Paper Formatting Concerns:**

no concerns

**Quality:**

4

**Strengths And Weaknesses:**

I am not very familiar with the existing literature, so I can't comment on the novelty of the result.

Strenghts

- The paper is very clearly written. It confirms superiority of Scout over alpha-beta as predicted in multiple numerical studies.

Weaknesses

- I believe the results are significant as-is, but they could be greatly improved if there was analysis of MCTS, as that is the leading algorithm for large deterministic games. The lack of MCTS analysis is pointed out in the related works section.

---

> ### Author Rebuttal · Authors · 2025-07-31
>
> We thank the reviewer for their positive assessment and insightful comments. We are glad they found the paper clearly written and the results significant.
>
> **On the Analysis of MCTS**
> >“I believe the results are significant as-is, but they could be greatly improved if there was analysis of MCTS, as that is the leading algorithm for large deterministic games. The lack of MCTS analysis is pointed out in the related works section.”
>
> We agree that an analysis of Monte-Carlo Tree Search (MCTS) would be a valuable and significant extension to this line of work. Our decision to focus on minimax-based algorithms in this paper was based on two primary considerations.
>
> First, while MCTS is indeed the state-of-the-art for many games, its theoretical guarantees differ from those of minimax algorithms. Algorithms like Alpha-Beta provide **deterministic guarantees** on finding the optimal move within a finite search. In contrast, MCTS algorithms like UCT offer **asymptotic convergence**, but their convergence rate can be extremely slow in certain worst-case scenarios (e.g., doubly exponential in the tree depth $h$), whereas minimax algorithms are at most exponential in $h$. See [1] or [2] for instance.
>
> Second, from a technical standpoint, the average-case analysis we perform is not straightforward to extend to MCTS. The recursive equations needed to characterize the complexity of UCT would be substantially more involved, as the algorithm's state depends on **visit counts and empirical rewards at each node**. While we believe such an analysis is possible in principle, it presents a formidable technical challenge that we view as an important direction for future research.
>
> **On Leaf-Disjoint Strategies**
> >“Does the level dependence in the *forward model* increase probability of leaf-disjoint strategies (as defined in [1]), compared to to the standard model?”
>
> This is an excellent question. We thank the reviewer for pointing us to this relevant work by Lorenz and Monien, which we were not previously aware of.
>
> As we understand it, Lorenz and Monien study the robustness of minimax search when applied with noisy leaf evaluations. Their key insight is that when multiple leaf-disjoint strategies (LDS) exist (i.e., alternative, non-overlapping paths that yield the same game-theoretic value) the algorithm becomes more robust to evaluation errors. In binary-valued games (Win = 1, Lose = 0), flipping a few outcomes is less likely to affect the minimax value if there are many such disjoint strategies.
>
> Our interpretation of a "leaf-disjoint strategy" is a set of moves that independently **proves** the value of the game. The existence of multiple such strategies (even in an exact solving setting) should make a game easier to solve for a minimax search algorithm, as a proof can be found more quickly, leading to more *aggressive pruning*.
>
> Applying this concept to our binary-valued *forward model* provides an interesting perspective on **game difficulty**:
>
> - When $q=1$ (the probability of a child being a 'loss' given the parent is a 'win'), any move for the MAX player is optimal. This corresponds to a maximum number of leaf-disjoint strategies ($b^{h/2}$), and it is precisely the easiest case for the algorithms we analyze, resulting in the smallest branching factor.
> - Conversely, when $q=0$, there is only one optimal move at each node for the MAX player. This results in a single leaf-disjoint strategy and represents the most difficult case for the search algorithm.
>
> This suggests that the number of leaf-disjoint strategies is **inversely correlated with the game's difficulty** for minimax search, mirroring the dynamics of the branching factor in our model.
>
> Regarding the *standard model,* the well-known value-collapse phenomenon means that for deep trees, game values tend to polarize. This behavior closely resembles the $q=1$ case in our forward model. We can therefore infer that the standard model tends to generate games with a very large number of leaf-disjoint strategies, at least for non-critical parameter settings.
>
> The connection between the number of leaf-disjoint strategies and minimax game-solving complexity looks promising and we believe there is something to investigate and to formalize here. We thank the reviewer again for highlighting this promising avenue for further investigation and framing the properties of our model in a new light.
>
> **References**
>
> [1] Coquelin, P. A., & Munos, R. (2007). Bandit algorithms for tree search. arXiv preprint cs/0703062.
>
> [2] Orseau, L., & Munos, R. (2024). Super-exponential regret for UCT, AlphaGo and variants. arXiv preprint arXiv:2405.04407.

---

> > ### Comment · Reviewer_8x4a · 2025-08-08
> >
> > Thank you for the clarification of difficulty of extending the results to MCTS.
> >
> > Glad you found the new reference useful. I believe your interpretation of it is correct and also your comments on relating it to the forward/standard model. It is not a well known work, but I think it's highly relevant for heuristic search with neural networks, which have inherently noisy evaluations of leaf nodes. I agree there is something to investigate and to formalize at this intersection.

---

### Official Review · Reviewer_j2Zu · 2025-07-03

**Clarity:** 3
**Significance:** 2
**Originality:** 3
**Rating:** 5
**Confidence:** 5

**Summary:**

This paper introduces a new method for constructing a virtual game tree and compares the standard Alpha–Beta algorithm with several alternatives, including SCOUT. The results confirm the empirically observed advantage of SCOUT over Alpha–Beta.

**Questions:**

I raised a question in the weaknesses section (second bullet). Could you please clarify this point so I can better understand your analysis?

**Ethical Concerns:**

["NO or VERY MINOR ethics concerns only"]

**Final Justification:**

I have raised my score based on the authors' rebuttal. My primary concerns about the paper's contribution and novelty have been addressed, as the authors have agreed to incorporate the suggested citations. Their response also cleared up some of my own misunderstandings.

**Limitations:**

Simply listing MTD(f) and PVS in the checklist is insufficient; a full discussion should be in the main text of the paper.

**Paper Formatting Concerns:**

Paper formatting is OK.

**Quality:**

3

**Strengths And Weaknesses:**

Strengths:
- The proposed "forward model" seems new.

Weaknesses:
- Lack of novelty. It has long been established that algorithms more efficient than Alpha–Beta exist—most notably MTD(f)—and that SCOUT generally outperforms the standard Alpha–Beta search. Demonstrating this well-known result again does not, by itself, constitute a new contribution.

- Unclear analysis in Section 5.1 (lines 178–180). The manuscript states:

> “… but under the assumption that the special child—which inherits the root’s value owing to the negamax formulation—has already been identified.”

I do not see how the **special child** can be identified **during** the search.
What exactly is being assumed in the subsequent analysis? For example, does the study first run a complete search to discover which move is the special child, and then use that information to analyse node counts? Please clarify the procedure and the underlying assumptions.

- Missing citations on forward-model tree generation. The paper does not reference earlier work that generates non-pathological game trees using techniques closely related to the forward model. A well-known example is the P-game described by Stephen J. J. Smith and Dana S. Nau, “An Analysis of Forward Pruning,” AAAI 1994. The same approach is also employed in Levente Kocsis and Csaba Szepesvári, “Bandit-Based Monte-Carlo Planning,” ECML 2006. Moreover, if the edge costs to child nodes are constrained to be non-positive, with the edge to the optimal move set to zero, the analysis becomes much simpler—a common trick whose original source, however, appears undocumented.

---

> ### Author Rebuttal · Authors · 2025-07-31
>
> We thank the reviewer for the review and the opportunity to clarify and expand on several points.
>
> **On the summary**
> >“This paper introduces a new method for constructing a virtual game tree and compares the standard Alpha–Beta algorithm with several alternatives, including SCOUT. The results confirm the empirically observed advantage of SCOUT over Alpha–Beta.”
>
> We believe that this summary doesn't fully capture the scope of our work. Our primary contributions are:
> 1. The introduction of a probabilistic **forward model** for generating random game trees that are richer and less pathological than those generated by the standard model commonly studied in the literature.
> 2. A formal, average-case complexity analysis of Solve, AlphaBeta, Scout, and Test algorithms **under this new model**. This involves deriving novel recursive equations, a closed-form for Solve and establishing a formal characterization of the branching factor for the other algorithms.
> 3. A numerical study on deep, finite trees generated under our model. Our analysis reveals that Scout consistently outperforms AlphaBeta across various game difficulties. This result, under a richer game generation model, is novel. To our knowledge, prior comparisons were limited to specific game instances or conducted under the standard model, which often produces degenerate trees.
>
> **On the lack of novelty**
>
> >“Lack of novelty. It has long been established that algorithms more efficient than Alpha–Beta exist—most notably MTD(f)—and that SCOUT generally outperforms the standard Alpha–Beta search. Demonstrating this well-known result again does not, by itself, constitute a new contribution.”
>
> We respectfully disagree that demonstrating this result is the sole contribution of our paper. As outlined above, our main contribution is the theoretical analysis of these algorithms under a new, richer game generation model. The conclusion that Scout outperforms Alpha-Beta is a *consequence* of this deeper analysis, not its starting premise. While it is true that Scout and other algorithms have long been known to outperform AlphaBeta in certain cases, the analyses supporting this were generally either small-scale and empirical, or restricted to the standard model, whose limitations we critique in the paper.
>
> **On the “unclear analysis”**
>
> >Unclear analysis in Section 5.1 (lines 178–180). The manuscript states:
> “… but under the assumption that the special child—which inherits the root’s value owing to the negamax formulation—has already been identified.”
> I do not see how the special child can be identified during the search. What exactly is being assumed in the subsequent analysis? For example, does the study first run a complete search to discover which move is the special child, and then use that information to analyze node counts? Please clarify the procedure and the underlying assumptions.
>
> There seems to be a misunderstanding here. None of the algorithms **ever** have access to intermediate node values prior to computing them from leaves: **algorithms only have access to leaf values**. The assumption mentioned is made purely for the purpose of the theoretical analysis, not as a feature of the algorithm. To derive the recursive equations for average-case complexity, we partition the analysis based on possible **probabilistic events**. We analyze the complexity conditioned on whether the "special child" (i.e., the one that will determine the parent's value) is the first one explored, the second one, and so on. By considering all possibilities and leveraging the properties of our forward model, we can formulate a recursive relationship for the expected number of nodes visited. This is a standard technique in the probabilistic analysis of algorithms.
>
> **On the “missing citations”**
>
> > Missing citations on forward-model tree generation. The paper does not reference earlier work that generates non-pathological game trees using techniques closely related to the forward model. A well-known example is the P-game described by Stephen J. J. Smith and Dana S. Nau, “An Analysis of Forward Pruning,” AAAI 1994.
>
> We believe there is a misunderstanding here. The P-game mentioned in Smith and Nau (1994) refers to the so-called **Pearl game** (named after Judea Pearl, see [1] for instance), which is exactly what we call the *standard model* throughout our paper. This model, based on i.i.d. leaf values, is indeed discussed extensively is the main text, and we chose to cite the relevant foundational work (including Pearl [2], and earlier work by Knuth & Moore [3] or Fuller [4]).
>
> > The same approach is also employed in Levente Kocsis and Csaba Szepesvári, Bandit-Based Monte-Carlo Planning,” ECML 2006.
>
> Regarding the Kocsis and Szepesvári (2006) paper, while the authors mention using P-games for evaluation, their description of reward calculation ("summing up the values along the path") is closer to what is sometimes called the incremental or N-game model, where values are assigned to edges and aggregated along the path. We discuss this line of work in the related work section, although we avoid using the P-game/N-game terminology since it is inconsistently applied in the literature.
> That said, we understand that this terminology may cause confusion and have added a footnote in the revised manuscript noting that the standard model is sometimes referred to as the P-game and the incremental model as the N-game.
>
> **On the analysis of MTD(f) and PVS**
>
> >Simply listing MTD(f) and PVS in the checklist is insufficient; a full discussion should be in the main text of the paper.
>
> We agree that the rationale for not including MTD(f) and PVS in our main analysis could be made clearer, as we only briefly discuss them in Section 6 and 7 (and the checklist). We provide the reasoning here and will include this explanation in the appendix of the revised manuscript.
>
> Our reasoning for excluding MTD(f) and PVS from the main analysis was primarily due to technical challenges in the derivation of the recursive complexity equations. The complexity of these algorithms depends not only on the parent node's value (as with AlphaBeta and Scout) but also on the specific certificate value returned by the TEST algorithm. This introduces an additional state variable that makes the recursive formulation significantly more difficult, and it was not clear if a closed-form analysis was feasible. In particular, while we control the nodes’ values with the distribution mu, we typically do not control the nodes’ test certificate values, it is plausible that their distributions can be computed and incorporated into the equations, but we didn’t manage to do it properly yet, and we leave this open for future work.
>
> Nevertheless, we believe our analysis provides strong proxies. Scout is closely related to PVS; the main difference is that PVS can sometimes use the certificate value from a TEST call to prune immediately, an optimization we believe has a minor impact on the asymptotic behavior. Similarly, our TEST-BISECTION algorithm is a simple variant of MTD(f), where the next search window is chosen by a bisection rule rather than by the certificate value from the previous search; here again we believe it shouldn’t impact the asymptotic behavior. Of course, this is only speculation, which is why we believe it’s important that PVS and MTD(f) are properly analyzed in the future.
>
> **References**
>
> [1] Nau, D. S. (1982). An investigation of the causes of pathology in games. Artificial Intelligence, 19(3), 257-278.
>
> [2] Pearl, J. (1980). Asymptotic properties of minimax trees and game-searching procedures. Artificial Intelligence, 14(2), 113-138.
>
> [3] Knuth, D. E., & Moore, R. W. (1975). An analysis of alpha-beta pruning. Artificial intelligence, 6(4), 293-326.
>
> [4] Fuller, S. H., Gaschnig, J. G., & Gillogly, J. J. (1973). Analysis of the alpha-beta pruning algorithm (No. AFOSRTR731549). Department of Computer Science, Carnegie-Mellon University.

---

> > ### Comment · Reviewer_j2Zu · 2025-08-08
> >
> > First, thank you for your clarification regarding the "unclear analysis." I believe I understand this point now and will adjust my score accordingly.
> >
> > Next, concerning the "lack of novelty." I must apologize that my initial comment was not specific enough. When I commented on the forward model, I had the "Prefix Value Game Trees" from the following paper in mind, but I failed to recall the specific reference when writing my initial review, which made my feedback incomplete.
> >
> > - Furtak, Timothy, and Michael Buro. "Minimum Proof Graphs and Fastest-Cut-First Search Heuristics." IJCAI. Vol. 9. 2009.
> >
> > Additionally, the concept of building artificial trees top-down has been previously explored, for example, in this paper:
> >
> > - Scheucher, Anton, and Hermann Kaindl. "Benefits of using multivalued functions for minimaxing." Artificial Intelligence 99.2 (1998): 187-208.
> >
> > The point I wish to emphasize is that the primary novelty of your work appears to lie in the analysis methodology and the resulting findings, rather than in the top-down construction of the artificial tree itself. As the manuscript is currently written, this distinction seems ambiguous.
> >
> > I am prepared to raise my score further if this point will be clarified in a revised version.

---

> > > ### Author Response · Authors · 2025-08-09
> > >
> > > Thank you for your reply to our rebuttal. We regret that it arrives so close to the end of the discussion period, leaving us with limited time to fully reflect on and elaborate upon the works you pointed out.
> > >
> > > We appreciate you directing us to these references. We were not aware of the existence of these works and we apologize for missing them in our literature review. We briefly summarize our understanding of these works:
> > >
> > > -   **[1]** The authors generate a game tree in a way similar to the incremental model, but at each node they sample a heuristic value and then deduce the true value from it, incorporating potential errors. This places the model between the incremental model and our forward model: it maintains heuristic values (like the incremental model) while also assigning true values to nodes (like our forward model).
> > >
> > > -   **[2]** The connection to our work is even clearer. The authors propose a **Prefix Value Game Tree Model** derived from the incremental (or N-game) model. By fixing the increment of one child per node to zero, the prefix value equals the negamax value. This resembles our forward model’s sampling step, where one child is assigned the negated value of the parent. This makes for a direct and useful link: the Prefix Value Game Tree Model can be seen as an additive variant of our forward model, connecting our approach back to the incremental model. We will ensure this work is properly cited in the revision and that the relationship is clearly explained.
> > >
> > > With regard to that, let's clarify our main contributions:
> > >
> > > 1.  A precisely specified formulation of a forward tree-generation model with a fixed conditional law at each node.  And the fact that this formulation is simple enough to allow a non-trivial theoretical analysis of classic algorithms.
> > >
> > > 2.  The analysis itself, i.e. the recursive equations for the average complexity of classic algorithms, along with the proofs of some interesting theoretical properties.
> > >
> > > 3.  The finite-depth numerical analysis, that uses the recursive equations to highlight practical performance discrepancies in the algorithms by looking at the sub-exponential multiplicative factor, for much larger trees than those feasible for Monte Carlo simulations.
> > >
> > > To the best of our knowledge, our work still provides **the first theoretical analysis of this conceptually simple yet rich probabilistic game model**, although now it is more clearly related to models discussed in the literature.
> > >
> > > We commit to duly reference [1] and [2] in the revised paper, and to clarify that the novelty lies in the exact formalization and analysis, rather than in the general idea of forward tree generation. We hope this addresses the reviewer's concerns.
> > >
> > >
> > > [1] Scheucher, Anton, and Hermann Kaindl. "Benefits of using multivalued functions for minimaxing." Artificial Intelligence 99.2 (1998): 187-208.
> > >
> > > [2] Furtak, Timothy, and Michael Buro. "Minimum Proof Graphs and Fastest-Cut-First Search Heuristics." IJCAI. Vol. 9. 2009

---

> > > > ### Comment · Reviewer_j2Zu · 2025-08-09
> > > >
> > > > First, let me once again apologize for pointing out the related work so close to the deadline.
> > > >
> > > > Thank you for agreeing to cite the papers and adjust your text; this fully addresses my concern. I also want to state that I am in full agreement with your claims about the paper's contribution.
> > > >
> > > > I will be revising my rating shortly to reflect our discussion.
> > > >
> > > > Thank you for your prompt and appropriate response, especially given the short notice.

---

### Official Review · Reviewer_aS2E · 2025-07-07

**Clarity:** 3
**Significance:** 2
**Originality:** 3
**Rating:** 3
**Confidence:** 3

**Summary:**

This paper critiques standard analyses of game-solving algorithms that assume game-tree leaf values are independent, arguing that this simplification makes problems unrealistically easy. To address this, the authors introduce a new probabilistic model that enforces ancestor dependency, better capturing the structure of real-world games while remaining analytically tractable. Using this model, they derive average-case complexity formulae for classical algorithms like AlphaBeta and Scout, enabling rigorous comparisons, even on deep trees where simulations fail.

Their key finding is that although all algorithms share the same asymptotic branching factor, finite-depth analysis reveals major differences: AlphaBeta suffers from a large multiplicative constant, slowing it down compared to alternatives like Scout. This suggests that prior claims of universal optimality were artifacts of an oversimplified model, and that more realistic frameworks are crucial for meaningful algorithm evaluation.

**Questions:**

54: Real-world games, from chess to Go, exhibit intricate dependencies between positions: the value of a leaf is inherently tied to the sequence of moves leading to it.

I'm not sure I follow this criticism, and it's not clear to me what independence means here.
It seems to me that in games like chess and go, it doesn't matter how a position is reached, once we are there, we are there. So in that sense, the value of a leaf is not a function of the sequence of operations that were used to reach it.
Are you referring to independence in a statistical sense? It seems later that you want to propose sampling in a way that the destination leaf depends on some structure higher up in the tree. This seems totally reasonable, but the term “independence” is still quite confusing to me. Exactly which sense did you mean? I suppose it would help me to clarify whether we are talking about mathematical or functional independence of a quantity on another quantity, as opposed to independence of random variables.

**Ethical Concerns:**

["NO or VERY MINOR ethics concerns only"]

**Final Justification:**

They definitely answered most of my questions, which I appreciate, but their clarifications don't really make me too much more excited about the paper. I don't dislike the paper, but I do feel like it's a bit thin on results, and in particular, I'm disappointed they couldn't use their framework to theoretically characterize the difference between alpha beta and scout.

**Limitations:**

One of the big claims is that for games with $n$ distinct utility values, alpha-beta search has a factor $O(n)$ overhead as compared to scout.

Specifically,
70: “However, finite-depth analysis, in Section 6, reveals critical practical differences masked by asymptotics. Specifically, alpha-beta incurs a multiplicative constant proportional to the game’s value range, causing significant slowdowns compared to other algorithms like scout.”

This makes it seem like a theorem is proved that shows that, in some sense, alpha-beta is less efficient than scout. Perhaps it's difficult, because we have to consider the asymptotics of multiple variables (depth and range), but I would like to see this theorem formally shown, rather than just plots and empirical results.


The discussion of a game's “value range”, beginning on line 72, is quite confusing, and notation seems to change between discrete {-n, …, n} and continuous [-n, n].

From the name, it sounds like we are talking about the difference between maximum and minimum value, which would be perfectly fine with continuous value sets. But then, on line 149, you use a discrete value range of 2n + 1 elements. If it’s inherently about discreteness, I’d use terms like “cardinality” or “size” instead of range. So far, this is section 5 rather than section 6, but section 5 doesn't yet distinguish between alpha-beta and scout.



Then in section 6, e.g., on line 314, we are no longer in the discrete world, as we sample continuous random variables. It seems like there is some discretization happening here in the test brute force and test by section algorithms, but it's not clear that alpha beta or scout are also being discretized.

It seems to me like with this discretization, these are actually approximation algorithms, and you implicitly set the approximation parameter to one. Is this the case? Are you identifying $n/\epsilon$ overhead? I'm reading a lot into the paper here, it would be very helpful if this were made explicit.

Theorem 5.1 also seems to mix continuous and discrete notation.

**Quality:**

2

**Strengths And Weaknesses:**

This paper re-examines how deterministic two-player zero-sum games, represented as uniform min/max trees, are used to evaluate game-solving algorithms. The authors introduce the forward model, which builds trees level-by-level using conditional distributions that preserve ancestor dependency. This better reflects the strategic coherence and the structural dependencies found in real games, and allows difficulty to be tuned.


This seems intuitive, but it is not always clear exactly how this model results in more realistic games. This is somewhat subjective, and more examples would help.

A big positive is that, using this model, they derive recursive formulas for the average-case complexity of algorithms like AlphaBeta and Scout. These formulas offer both asymptotic insight and tractable simulation of deep trees (up to height ~5000), unlike expensive Monte Carlo methods. However, I'd like more precise theoretical characterization, ideally in terms of distribution properties and abstract characterizations of games under their model, rather than empirical plots of limited ranges with specific distributions.

---

> ### Author Rebuttal · Authors · 2025-07-30
>
> We thank the reviewer for the thoughtful comments and valuable suggestions, which helped us clarify and improve the presentation of our work.
>
> **On the realism of the forward model**
>
> > “This seems intuitive, but it is not always clear exactly how this model results in more realistic games. This is somewhat subjective, and more examples would help.”
>
> We appreciate this feedback. To clarify, our intent was not to claim that the forward model generates realistic games in an **absolute sense**, but rather that it generates instances that are richer and more challenging than those produced by the *standard model*. The standard model, as we discuss, exhibits several *pathological* simplifications: (1) root values are deterministic in almost all cases, (2) all algorithms solve these instances in globally optimal time outside of the critical case, and (3) even in the critical case, the games remain not maximally hard.
>
> In contrast, the *forward model* introduces controllable randomness and structure by design. Specifically, root values become genuinely stochastic (tuned by the input distribution $\mu$), and we are able to span the **full difficulty spectrum**, from trivial to maximally hard games. Given that this terminology (“more realistic”) caused confusion among reviewers, we revised the manuscript to replace it with “richer” in the relevant instances.
>
>
> **On obtaining more precise characterization of the complexity of the different algorithms**
>
> > “However, I'd like more precise theoretical characterization, ideally in terms of distribution properties and abstract characterizations of games under their model, rather than empirical plots of limited ranges with specific distributions.”
>
> We fully agree with the reviewer that deeper theoretical characterization would be valuable. In the binary-valued setting, we were indeed able to obtain a **closed-form** expression for average-case complexity and branching factor (see Section 4 and Appendix C). However, for more general discrete-valued trees, such closed-form solutions remain out of reach for us.
>
> Nevertheless, we provide a characterization of the branching factor $r$ as the **spectral radius** of a matrix associated with our recursive system (see Section 5). While this does not yield a closed-form expression for arbitrary $\mu$, it allows efficient and precise numerical evaluation over extremely deep trees, far beyond what Monte Carlo methods can achieve.
>
> To give an idea of why it may be harder to obtain a closed form in the forward model than in the standard model, we point out that in the standard model the branching factor **can only take two values**. Regardless of the exact distribution $\mu$, what only matters is whether $\mu$ is critical or not. In our model, and it is very clear in the binary-valued case, the branching factor depends continuously on the distribution (Section 4, Equation 5), so it can take an **infinite number of values** between theoretical lower and upper bounds.
>
>
> **On the independence of leaf values.**
>
> > “I'm not sure I follow this criticism, and it's not clear to me what independence means here. It seems to me that in games like chess and go, it doesn't matter how a position is reached, once we are there, we are there.”
>
> Yes, we are referring specifically to **statistical** independence of leaf values, as assumed in the standard model. The critique is not that the value of a state depends on the path in a functional sense (e.g., due to move repetition), but rather that in real games, the distribution of terminal outcomes in a subtree is strongly shaped by the position from which it originates.
>
> For instance, in chess, if a player is a queen up in a given position, we expect the downstream leaves to overwhelmingly favor that player (see [1] and [2] for similar examples). This reflects a statistical correlation between the position and the outcomes that follow from it. The standard model breaks this structure by assuming independent sampling at the leaves, leading to unrealistic and overly uniform game trees. Our forward model aims to reintroduce this type of correlation, while remaining simple enough to allow a **tractable analysis**.
>
>
> **On the confusion of discrete vs continuous variables.**
>
> > “The discussion of a game's “value range”, beginning on line 72, is quite confusing, and notation seems to change between discrete {-n, …, n} and continuous [-n, n].”
>
> We thank the reviewer for pointing this out. All occurrences of “[-n, n]” were indeed **typographical errors**; our model and experiments use only discrete values from the set {-n, ..., n}. These have now been corrected throughout the manuscript.
>
> We chose to work in the discrete setting primarily for clarity and analytical tractability. As noted in response to *Reviewer cYDt*, extending the model to continuous values is conceptually straightforward (replacing sums with integrals in the recursive equations), but significantly complicates formal results like Theorem 5.1. Furthermore, most classical games of interest (e.g., chess, Go) have discrete outcomes, justifying this choice for the scope of our study.
>
> Regarding the approximation concern: the algorithms we study (AlphaBeta, Scout, etc.) are not approximating the value of the game. The discretization is not an approximation artifact but rather an inherent property of the value space used in our experiments.
>
> **References**
>
> [1] Nau, D. S. (1982). An investigation of the causes of pathology in games. Artificial Intelligence, 19(3), 257-278.
>
> [2] Knuth, D. E., & Moore, R. W. (1975). An analysis of alpha-beta pruning. Artificial intelligence, 6(4), 293-326.

---

> > ### Comment · Reviewer_aS2E · 2025-08-06
> >
> > I thank the authors for their clarifications. I see now that the results are inherently about cardinality, rather than numeric ranges per se, but I still don't understand why those cardinalities couldn't be incorporated into the asymptotic runtime analysis of Scout and Alpha-Beta. It seems like this is a natural parameter that these algorithms' runtimes would depend on, and you found a linear difference in their performance, but why does this not appear in the analysis?

---

> > > ### Author Response · Authors · 2025-08-06
> > >
> > > Thank you for engaging in the discussion with us. If we understand your question correctly, you are asking why the cardinality of the value set, determined by the parameter $n$, does not explicitly appear in our asymptotic runtime analysis.
> > >
> > > We clarify that it does appear, but is **implicitly encoded** in the recursive equations that define the algorithm’s behavior. Specifically, the average-case complexity of an algorithm $A$ is expressed as:
> > >
> > > $I_{A} =  K(b,n,\mu, h) r(b,n,\mu)^h+ o( r(b,n,\mu)^h)$
> > >
> > > where $r$ is the branching factor that determines the exponential rate. This branching factor is computed as the **spectral radius** of a matrix $M_A$ derived from the recursive equations: $r = \rho(M_A)$.
> > >
> > > The matrix $M_A$, and therefore $r$, depends on $n$ and the distribution $\mu$. We prove that AlphaBeta and Test share the same branching factor in Section 5, and experimental evidence suggests Scout does too.
> > >
> > > Since all branching factor coincide, it is interesting to look at the multiplicative factor $K(b,n,μ,h)$ for the different algorithms. Even though it doesn't impact the asymptotic branching factor, as it gets squashed by exponentiation to $1/h$, it's informative on the practical speed of the algorithms on large games. In Figure 2 of the paper, we normalize the cost of each algorithm by the cost of Average-Test, removing the exponential component $r^h$ and isolating the multiplicative ratio: $\phi_A = K_A / K_{Avg-test}$. This ratio reflects relative efficiency. Empirically, we observed that for large $h$, $\phi_{\text{AlphaBeta}}$ grows roughly linearly with $n$, while $\phi_{\text{Scout}}$ grows more slowly.
> > >
> > > Unfortunately, we didn't manage to deduce a precise characterization or closed form for the multiplicative factor $K$ nor for the ratio $\phi$  , so we only have experimental results to build upon.
> > >
> > >  To further support this, we conducted new experiments fixing $b = 10$,  $h=100$ (in our experiments $h=100$ is enough to get to the asymptotic regime) and using a uniform distribution over values. Below is the result, comparing $\phi$ as a function of $n$:
> > >
> > >
> > >
> > >
> > >
> > > |$n$|$\phi_{AlphaBeta}$| $\phi_{Scout}$| $2n$ | $1+\log{2n}$|
> > > |----:|---------:|--------:|-----:|----------:|
> > > | 1 | 2.00 | 1.41| 2.00 | 1.69 |
> > > | 2 | 4.00 | 1.91 | 4.00 | 2.39 |
> > > | 3 | 6.00 | 2.33 | 6.00 | 2.79 |
> > > | 4 | 7.00 | 2.66 | 8.00 | 3.08 |
> > > | 5 | 10.00 | 2.93 | 10.00 | 3.30 |
> > > | 6 | 12.00 | 3.16 | 12.00 | 3.48 |
> > > | 7 | 13.00 | 3.36 | 14.00 | 3.64 |
> > > | 8 | 15.99 | 3.53 | 16.00 | 3.77 |
> > > | 9 | 17.97 | 3.68 | 18.00 | 3.89 |
> > > | 10 | 19.95 | 3.81 | 20.00 | 3.99 |
> > >
> > > These results suggest that AlphaBeta’s runtime overhead scales roughly as $\mathcal{O}(n)$, while Scout’s is closer to $\mathcal{O}(\log n)$ in practice. This pattern held consistently across multiple distributions and branching factors (not shown here for brevity).
> > >
> > > We hope this clarifies how $n$ appears in the analysis, both in the asymptotic analysis through the spectral radius of a matrix, and in the numerical experiments via the multiplicative constants. Please let us know if this matches your interpretation, and whether it addresses your concern. We're happy to incorporate further clarification in the final version.

---

### Decision · Program_Chairs · 2025-09-17

**Decision:**

Accept (poster)

**Comment:**

This paper addresses a fundamental issue in the average-case analysis of game-solving algorithms, specifically the assumption that game-tree leaf values are independent. The authors argue that this simplification renders problems unrealistically easy. To tackle this issue, they propose the "forward model," a new probabilistic framework that enforces ancestor dependency. This model more accurately reflects the structure of real-world games while remaining analytically tractable. Using this framework, the authors derive average-case complexity formulas for classical algorithms such as Alpha-Beta and Scout, enabling rigorous comparisons even in deep game trees where simulations typically fail.

The paper has received five reviews with scores ranging from weak rejection to clear acceptance. All reviewers recognize the novelty of the forward model, the rigorous analysis of deterministic strategies, and the clarity of the writing. However, there is debate over the theoretical significance of the results; a formal characterization of the advantages of Scout compared to Alpha-Beta would further highlight the strengths of this forward-model analysis.

Overall, the strengths of this paper outweigh its weaknesses, and I recommend acceptance. Still, I suggest taking into account the constructive discussions during the rebuttal phase for further revisions.